# Theory and Evaluation Metrics for Learning Disentangled Representations

**Kien Do and Truyen Tran**
Applied AI Institute, Deakin University, Geelong, Australia
{dkdo,truyen.tran}@deakin.edu.au

## Abstract

We make two theoretical contributions to disentanglement learning by (a) defining precise semantics of disentangled representations, and (b) establishing robust metrics for evaluation. First, we characterize the concept "disentangled representations" used in supervised and unsupervised methods along three dimensions–*informativeness*, *separability* and *interpretability*–which can be expressed and quantified explicitly using information-theoretic constructs. This helps explain the behaviors of several well-known disentanglement learning models. We then propose robust metrics for measuring informativeness, separability, and interpretability. Through a comprehensive suite of experiments, we show that our metrics correctly characterize the representations learned by different methods and are consistent with qualitative (visual) results. Thus, the metrics allow disentanglement learning methods to be compared on a fair ground. We also empirically uncovered new interesting properties of VAE-based methods and interpreted them with our formulation. These findings are promising and hopefully will encourage the design of more theoretically driven models for learning disentangled representations.

## 1 Introduction

Disentanglement learning holds the key for understanding the world from observations, transferring knowledge across different tasks and domains, generating novel designs, and learning compositional concepts (Bengio et al., 2013; Higgins et al., 2017b; Lake et al., 2017; Peters et al., 2017; Schmidhuber, 1992). Assuming the observation $x$ is generated from latent factors $z$ via $p(x|z)$, the goal of disentanglement learning is to correctly uncover a set of independent factors $\{z_i\}$ that give rise to the observation. While there has been a considerable progress in recent years, common assumptions about disentangled representations appear to be inadequate (Locatello et al., 2019).

Unsupervised disentangling methods are highly desirable as they assume no prior knowledge about the ground truth factors. These methods typically impose constraints to encourage independence among latent variables. Examples of constraints include forcing the variational posterior $q(z|x)$ to be similar to a factorial $p(z)$ (Burgess et al., 2018; Higgins et al., 2017a), forcing the variational aggregated prior $q(z)$ to be similar to the prior $p(z)$ (Makhzani et al., 2015), adding total correlation loss (Kim & Mnih, 2018), forcing the covariance matrix of $q(z)$ to be close to the identity matrix (Kumar et al., 2017), and using a kernel-based measure of independence (Lopez et al., 2018). However, it remains unclear how the independence constraint affects other properties of representation. Indeed, more independence may lead to higher reconstruction error in some models (Higgins et al., 2017a; Kim & Mnih, 2018). Worse still, the independent representations may mismatch human's predefined concepts (Locatello et al., 2019). This suggests that supervised methods – which associate a representation (or a group of representations) $z_i$ with a particular ground truth factor $y_k$ – may be more adequate. However, most supervised methods have only been shown to perform well on toy datasets (Harsh Jha et al., 2018; Kulkarni et al., 2015; Mathieu et al., 2016) in which data are generated from multiplicative combination of the ground truth factors. It is still unclear about their performance on real datasets.

We believe that there are at least two major reasons for the current unsatisfying state of disentanglement learning: i) the lack of a formal notion of disentangled representations to support the design of proper objective functions (Tschannen et al., 2018; Locatello et al., 2019), and ii) the lack of robust evaluation metrics to enable a fair comparison between models, regardless of their architectures or

design purposes. To that end, we contribute by formally characterizing disentangled representations along three dimensions, namely *informativeness*, *separability* and *interpretability*, drawing from concepts in information theory (Section 2). We then design robust quantitative metrics for these properties and argue that an ideal method for disentanglement learning should achieve high performance on these metrics (Section 3).

We run a series of experiments to demonstrate how to compare different models using our proposed metrics, showing that the quantitative results provided by these metrics are consistent with visual results (Section 4). In the process, we gain important insights about some well-known disentanglement learning methods namely FactorVAE (Kim & Mnih, 2018), $\beta$-VAE (Higgins et al., 2017a), and AAE (Makhzani et al., 2015).

## 2 RETHINKING DISENTANGLEMENT

Inspired by (Bengio et al., 2013; Ridgeway, 2016), we adopt the notion of disentangled representation learning as "a process of *decorrelating information* in the data into separate *informative* representations, each of which corresponds to a concept *defined by humans*". This suggests three important properties of a disentangled representation: *informativeness*, *separability* and *interpretability*, which we quantify as follows:

**Informativeness**   We formulate the *informativeness* of a particular representation (or a group of representations) $z_i$ w.r.t. the data $x$ as the mutual information between $z_i$ and $x$:

$$I(x, z_i) = \int_x \int_z p_{\mathcal{D}}(x) q(z_i|x) \log \frac{q(z_i|x)}{q(z_i)} \, dz \, dx \tag{1}$$

where $q(z_i) = \int_x p_{\mathcal{D}}(x) q(z_i|x) \, dx$. In order to represent the data faithfully, a representation $z_i$ should be informative of $x$, meaning $I(x, z_i)$ should be large. Because $I(x, z_i) = H(z_i) - H(z_i|x)$, a large value of $I(x, z_i)$ means that $H(z_i|x) \approx 0$ given that $H(z_i)$ can be chosen to be relatively fixed. In other words, if $z_i$ is informative w.r.t. $x$, $q(z_i|x)$ *usually has small variance*. It is important to note that $I(x, z_i)$ in Eq. 1 is defined on the variational encoder $q(z_i|x)$, and does not require a decoder. It implies that we do not need to minimize the reconstruction error over $x$ (e.g., in VAEs) to increase the informativeness of a particular $z_i$.

**Separability and Independence**   Two representations $z_i$, $z_j$ are *separable* w.r.t. the data $x$ if they do not share common information about $x$, which can be formulated as follows:

$$I(x, z_i, z_j) = 0 \tag{2}$$

where $I(x, z_i, z_j)$ denotes the multivariate mutual information (McGill, 1954) between $x$, $z_i$ and $z_j$. $I(x, z_i, z_j)$ can be decomposed into standard bivariate mutual information terms as follows:

$$I(x, z_i, z_j) \quad = \quad I(x, z_i) + I(x, z_j) - I(x, (z_i, z_j)) = I(z_i, z_j) - I(z_i, z_j|x)$$

$I(x, z_i, z_j)$ can be either positive or negative. It is positive if $z_i$ and $z_j$ contain redundant information about $x$. The meaning of a negative $I(x, z_i, z_j)$ remains elusive (Bell, 2003).

Achieving separability with respect to $x$ does not guarantee that $z_i$ and $z_j$ are separable in general. $z_i$ and $z_j$ are *fully separable* or *statistically independent* if and only if:

$$I(z_i, z_j) = 0 \tag{3}$$

If we have access to all representations $z$, we can generally say that a representation $z_i$ is *fully separable* (from other representations $z_{\neq i}$) if and only if $I(z_i, z_{\neq i}) = 0$.

Note that there is a trade-off between informativeness, independence and the number of latent variables which we discuss in Appdx. A.7.

**Interpretability**   Obtaining informative and independent representations does not guarantee interpretability by human (Locatello et al., 2019). We argue that in order to achieve interpretability, we should provide models with a set of predefined concepts $y$. In this case, a representation $z_i$ is

interpretable with respect to $y_k$ if it only contains information about $y_k$ (given that $z_i$ is separable from all other $z_{\neq i}$ and all $y_k$ are distinct). *Full interpretability* can be formulated as follows:

$$I(z_i, y_k) = H(z_i) = H(y_k) \tag{4}$$

Eq. 4 is equivalent to the condition that $z_i$ is an *invertible function* of $y_k$. If we want $z_i$ to generalize beyond the observed $y_k$ (i.e., $H(z_i) > H(y_k)$), we can change the condition in Eq. 4 into:

$$I(z_i, y_k) = H(y_k) \quad \text{or} \quad H(y_k|z_i) = 0 \tag{5}$$

which suggests that the model should accurately predict $y_k$ given $z_i$. If $z_i$ satisfies the condition in Eq. 5, it is said to be *partially interpretable* w.r.t $y_k$.

In real data, underlying factors of variation are usually correlated. For example, men usually have beard and short hair. Therefore, it is very difficult to match independent latent variables to different ground truth factors at the same time. We believe that in order to achieve good interpretability, we should isolate the factors and learn one at a time.

## 2.1 AN INFORMATION-THEORETIC DEFINITION OF DISENTANGLED REPRESENTATIONS

Given a dataset $\mathcal{D} = \{x_i\}_{i=1}^N$, where each data point $x$ is associated with a set of $K$ labeled factors of variation $y = \{y_1, ..., y_K\}$. Assume that there exists a mapping from $x$ to $m$ groups of latent representations $z = \{z_1, z_2, ..., z_m\}$ which follows the distribution $q(z|x)$. Denoting $q(z_i|x) = \sum_{z_{\neq i}} q(z|x)$ and $q(z_i) = \mathbb{E}_{p_{\mathcal{D}}(x)}[q(z_i|x)]$. We define disentangled representations for **unsupervised cases** as follows:

**Definition 1** (Unsupervised). A representation or a group of representations $z_i$ is said to be *"fully disentangled"* w.r.t a ground truth factor $y_k$ if $z_i$ is *fully separable* (from $z_{\neq i}$) and $z_i$ is *fully interpretable* w.r.t $y_k$. Mathematically, this can be written as:

$$I(z_i, z_{\neq i}) = 0 \quad \text{and} \quad I(z_i, y_k) = H(z_i, y_k) \tag{6}$$

The definition of disentangled representations for **supervised cases** is similar as above except that now we model $q(z|x, y)$ instead of $q(z|x)$ and $q(z) = \sum_{x,y} p_{\mathcal{D}}(x, y)q(z|x, y)$.

Recently, there have been several works (Eastwood & Williams, 2018; Higgins et al., 2018; Ridgeway & Mozer, 2018) that attempted to define disentangled representations. Higgin et. al. (Higgins et al., 2018) proposed a definition based on group theory (Cohen & Welling, 2014) which is (informally) stated as follows: "A representation $z$ is disentangled w.r.t a particular subgroup $y_k$ (from a symmetry group $y = \{y_k\}_{k=1}^K$) if $z$ can be decomposed into different subspaces $\{z_i\}_{i=1}^H$ in which the subspace $z_i$ should be independent of all other representation subspaces $z_{\neq i}$, and $z_i$ should only be affected by the action of a single subgroup $y_k$ and not by other subgroups $y_{\neq k}$.". Their definition shares similar observation as ours. However, it is less convenient for designing models and metrics than our information-theoretic definition.

Eastwood et. al. (Eastwood & Williams, 2018) did not provide any explicit definition of disentangled representations but characterizing them along three dimensions namely *"disentanglement"*, *"compactness"*, and *"informativeness"* (between $z$ any $y_k$). A high "disentanglement" score ($\approx 1$) for $z_i$ indicates that it captures at most one factor, let's say $y_k$. A high "completeness" score ($\approx 1$) for $y_k$ indicates that it is captured by at most one latent $z_j$ and $j$ is likely to be $i$. A high "informativeness" score[1] for $y_k$ indicates that all information of $y_k$ is captured by the representations $z$. Intuitively, when all the three notions achieve optimal values, there should be only a single representation $z_i$ that captures all information of the factor $y_k$ but no information from other factors $y_{\neq k}$. However, even in that case, $z_i$ is still *not* fully interpretable w.r.t $y_k$ since $z_i$ may contain some information in $x$ that does not appear in $y_k$. This makes their notions only applicable to toy datasets on which we know that the data $x$ are only generated from predefined ground truth factors $y = \{y_k\}_{k=1}^K$. Our definition can handle the situation where we only know some but not all factors of variation in the data. The notions in (Ridgeway & Mozer, 2018) follow those in (Eastwood & Williams, 2018), hence, suffer from the same disadvantage.

---

[1] In (Eastwood & Williams, 2018), the authors consider the prediction error of $y_k$ given $z$ instead. High "informativeness" score means this error should be close to 0.

## 3 ROBUST EVALUATION METRICS

We argue that a robust metric for disentanglement should meet the following criteria: i) it supports both supervised/unsupervised models; ii) it can be applied for real datasets; iii) it is computationally straightforward, i.e. not requiring any training procedure; iv) it provides consistent results across different methods and different latent representations; and v) it agrees with qualitative (visual) results. Here we propose information-theoretic metrics to measure informativeness, independence and interpretability which meet all of these robustness criteria.

### 3.1 METRICS FOR INFORMATIVENESS

We measure the informativeness of a particular representation $z_i$ w.r.t. $x$ by computing $I(x, z_i)$. If $z_i$ is discrete, we can compute $I(x, z_i)$ exactly by using Eq. 1 but with the integral replaced by the sum. If $z_i$ is continuous, we estimate $I(x, z_i)$ via sampling or quantization. Details about these estimations are provided in Appdx. A.10.

If $H(z_i)$ is estimated via quantization, we will have $0 \leq I(x, z_i) \leq H(z_i)$. In this case, we can divide $I(x, z_i)$ by $H(z_i)$ to normalize it to the range $[0, 1]$. However, this normalization may change the interpretation of the metric and lead to a situation where a representation $z_i$ is less informative than $z_j$ (i.e., $I(x, z_i) < I(x, z_j)$) but still has a higher rank than $z_j$ because $H(z_i) < H(z_j)$. A better way is to divide $I(x, z_i)$ by $\log(\#\text{bins})$.

### 3.2 METRICS FOR SEPARABILITY AND INDEPENDENCE

**MISJED** We can characterize the independence between two latent variables $z_i$, $z_j$ based on $I(z_i, z_j)$. However, a serious problem of $I(z_i, z_j)$ is that it generates the following order among pairs of representations:
$$I(z_{\text{f},i}, z_{\text{f},j}) > I(z_{\text{f},i}, z_{\text{n},j}) > I(z_{\text{n},i}, z_{\text{n},j}) \geq 0$$
where $z_{\text{f},i}$, $z_{\text{f},j}$ are informative representations and $z_{\text{n},i}$, $z_{\text{n},j}$ are uninformative (or noisy) representations. This means if we simply want $z_i$, $z_j$ to be independent, the best scenario is that *both are noisy and independent* (e.g. $q(z_i|x) \approx q(z_j|x) \approx \mathcal{N}(0, \text{I})$). Therefore, we propose a new metric for independence named **MISJED** (which stands for Mutual Information Sums Joint Entropy Difference), defined as follows:

$$
\begin{aligned}
\text{MISJED}(z_i, z_j) = \tilde{I}(z_i, z_j) &= H(z_i) + H(z_j) - H(\bar{z}_i, \bar{z}_j) \\
&= H(z_i) + H(z_j) - H(z_i, z_j) + H(z_i, z_j) - H(\bar{z}_i, \bar{z}_j) \\
&= I(z_i, z_j) + H(z_i, z_j) - H(\bar{z}_i, \bar{z}_j)
\end{aligned}
\tag{7}
$$

where $\bar{z}_i = \mathbb{E}_{q(z_i|x)}[z_i]$ and $q(\bar{z}_i) = \mathbb{E}_{p_{\mathcal{D}}(x)}[q(\bar{z}_i|x)]$. Since $q(\bar{z}_i)$ and $q(\bar{z}_j)$ have less variance than $q(z_i)$ and $q(z_j)$, respectively, $H(z_i, z_j) - H(\bar{z}_i, \bar{z}_j) \geq 0$, making $\tilde{I}(z_i, z_j) \geq 0$.

To achieve a small value of $\tilde{I}(z_i, z_j)$, two representations $z_i$, $z_j$ should be both *independent* and *informative* (or, in an extreme case, are deterministic given $x$). Using the MISJED metric, we can ensure the following order: $0 \leq \tilde{I}(z_{\text{f},i}, z_{\text{f},j}) < \tilde{I}(z_{\text{f},i}, z_{\text{n},j}) < \tilde{I}(z_{\text{n},i}, z_{\text{n},j})$. If $H(z_i)$, $H(z_j)$, and $H(\bar{z}_i, \bar{z}_j)$ in Eq. 7 are estimated via quantization, we will have $\tilde{I}(z_i, z_j) \leq H(z_i) + H(z_j) \leq 2\log(\#\text{bins})$. In this case, we can divide $\tilde{I}(z_i, z_j)$ by $2\log(\#\text{bins})$ to normalize it to $[0, 1]$.

**WSEPIN and WINDIN** A theoretically correct way to verify that a particular representation $z_i$ is both *separable* from other $z_{\neq i}$ and *informative* w.r.t $x$ is considering the amount of information in $x$ but not in $z_{\neq i}$ that $z_i$ contains. This quantity is the conditional mutual information between $x$ and $z_i$ given $z_{\neq i}$, which can be decomposed as follows:

$$
\begin{aligned}
I(x, z_i|z_{\neq i}) &= I(x, z_i) - I(x, z_i, z_{\neq i}) \\
&= I(x, z_i) - (I(z_i, z_{\neq i}) - I(z_i, z_{\neq i}|x)) \\
&= I(x, z_i) - I(z_i, z_{\neq i}) + I(z_i, z_{\neq i}|x)
\end{aligned}
\tag{8}
$$

$I(x, z_i|z_{\neq i})$ is useful for measuring how disentangled a representation $z_i$ is *in the absence of ground truth factors*. $I(x, z_i|z_{\neq i})$ is close to 0 if $z_i$ is completely noisy and is high if $z_i$ is disentangled[2].

---

[2]Note that only informativeness and separability are considered in this case.

For models that use factorized encoders, $z_i$ and $z_{\neq i}$ are usually assumed to be independent given $x$, hence, $I(z_i, z_{\neq i}|x) \approx 0$ and $I(x, z_i|z_{\neq i}) \approx I(x, z_i) - I(z_i, z_{\neq i})$ which is merely the difference between the *informativeness* and *full separability* of $z_i$. For models that use auto-regressive encoders, $I(z_i, z_{\neq i}|x) > 0$ which means $z_i$ and $z_{\neq i}$ can share information *not* in $x$.

We can also compute $I(x, z_i|z_{\neq i})$ in a different way as follows:

$$I(x, z_i|z_{\neq i}) = I(x, (z_i, z_{\neq i})) - I(x, z_{\neq i})$$
$$= I(x, z) - I(x, z_{\neq i})$$

If we want $z_i$ to be both *independence* of $z_{\neq i}$ and informative w.r.t $x$, we can only use the first two terms in Eq. 8 to derive another quantitive measure:

$$\hat{I}(x, z_i|z_{\neq i}) = I(x, z_i) - I(z_i, z_{\neq i})$$
$$= I(x, z_i|z_{\neq i}) - I(z_i, z_{\neq i}|x) \tag{9}$$

However, unlike $I(x, z_i|z_{\neq i})$, $\hat{I}(x, z_i|z_{\neq i})$ can be negative.

To normalize $I(x, z_i|z_{\neq i})$, we divide it by $H(z_i)$ ($H(z_i)$ must be estimated via quantization). Note that taking the average of $I(z_i, x|z_{\neq i})$ over all representations to derive a single metric for the whole model is *not appropriate* because *models with more noisy latent variables will be less favored*. For example, if model A has 10 latent variables (5 of them are disentangled and 5 of them are noisy), and model B has 20 latent variables (5 of them are disentangled and 15 of them are noisy), B will always be considered worse than A *despite the fact that both are equivalent* in term of disentanglement (since 5 disentangled representations are enough to capture all information in $x$ so additional latent variables should be noisy). We propose two solutions for this issue. In the first approach, we sort $I(x, z_i|z_{\neq i})$ over all representations in descending order and only take the average over the top $k$ latents (or groups of latents). This leads to a metric called SEPIN@$k$[3] which is similar to Precision@$k$:

$$\text{SEPIN@}k = \frac{1}{k}\sum_{i=0}^{k-1} I(x, z_{r_i}|z_{\neq r_i})$$

where $r_1, ..., r_L$ is the rank indices of $L$ latent variables by sorting $I(x, z_i|z_{\neq i})$ ($i = 1, ..., L$).

In the second approach, we compute the average over all $L$ representations $z_0, ..., z_{L-1}$ but weighted by their informativeness to derive a metric called WSEPIN:

$$\text{WSEPIN} = \sum_{i=0}^{L-1} \rho_i I(x, z_i|z_{\neq i})$$

where $\rho_i = \frac{I(x, z_i)}{\sum_{j=0}^{L-1} I(x, z_j)}$. If $z_i$ is a noisy representation, $I(x, z_i) \approx 0$, thus, $z_i$ contributes almost nothing to the final WSEPIN.

Similarly, using the measure $\hat{I}(x, z_i|z_{\neq i})$ in Eq. 9, we can derive other two metrics INDIN@$k$[4] and WINDIN as follows:

$$\text{INDIN@}k = \frac{1}{k}\sum_{i=0}^{k-1} \hat{I}(x, z_{r_i}|z_{\neq r_i}) \quad \text{and} \quad \text{WINDIN} = \sum_{i=0}^{L-1} \rho_i \hat{I}(x, z_i|z_{\neq i})$$

### 3.3 METRICS FOR INTERPRETABILITY

Recently, several metrics have been proposed to quantitatively evaluate the interpretability of representations by examining the relationship between the representations and manually labeled factors of variation. The most popular ones are Z-diff score (Higgins et al., 2017a; Kim & Mnih, 2018), SAP (Kumar et al., 2017), MIG (Chen et al., 2018). Among them, only MIG is theoretically sound and provides correct computation of $I(x, z_i)$. MIG also matches with our formulation of "interpretability" in Section 2 to some extent. However, MIG has only been used for toy datasets

---

[3] SEPIN stands for SEParability and INformativeness
[4] INDIN stands for INDependence and INformativeness

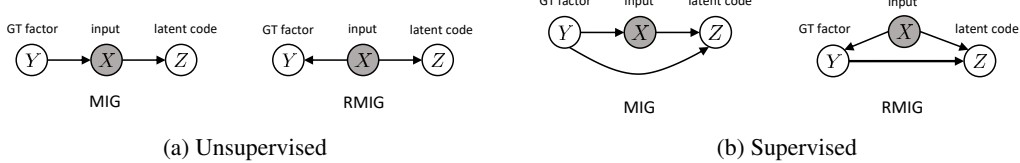

(a) Unsupervised            (b) Supervised

Figure 1: Differences in probabilistic assumption of MIG and Robust MIG.

like dSprites (Matthey et al., 2017). The main drawback comes from its probabilistic assumption $p(z_i, y_k, x^{(n)}) = q(z_i|x^{(n)})p(x^{(n)}|y_k)p(y_k)$ (see Fig. 1). Note that $p(x^{(n)}|y_k)$ is a distribution over the high dimensional data space, and is very hard to robustly estimate but the authors simplified it to be $p(n|y_k)$ if $x^{(n)} \in \mathcal{D}_{y_k}$ ($\mathcal{D}_{y_k}$ is the support set for a particular value $y_k$) and 0 otherwise. This equation only holds for toy datasets where we know exactly how $x$ is generated from $y$. In addition, since $p(n|y_k)$ depends on the value of $y_k$, it will be problematic if $y_k$ is continuous.

**RMIG** Addressing the drawbacks of MIG, we propose RMIG (which stands for Robust MIG), formulated as follows:

$$\text{RMIG}(y_k) \quad = \quad I(z_{i^*}, y_k) - I(z_{j^\circ}, y_k) \tag{10}$$

where $I(z_{i^*}, y_k)$ and $I(z_{j^\circ}, y_k)$ are the *highest* and the *second highest* mutual information values computed between every $z_i$ and $y_k$; $z_{i^*}$ and $z_{j^\circ}$ are the corresponding latent variables. Like MIG, we can normalize RMIG$(y_k)$ to [0, 1] by dividing it by $H(y_k)$ but it will favor imbalanced factors (small $H(y_k)$).

RMIG inherits the idea of MIG but differs in the probabilistic assumption (and other technicalities). RMIG assumes that $p(z_i, y_k, x^{(n)}) = q(z_i|x^{(n)})p(y_k|x^{(n)})p(x^{(n)})$ for unsupervised learning and $p(z_i, y_k, x^{(n)}) = q(z_i|y_k^{(n)}, x^{(n)})p(y_k^{(n)}, x^{(n)})$ for supervised learning (see Fig. 1). Not only this eliminates all the problems of MIG but also provides additional advantages. First, we can estimate $q(z_i, y_k)$ using Monte Carlo sampling on $p(x^{(n)})$. Second, $p(y_k|x^{(n)})$ is well defined for both discrete/continuous $y_k$ and deterministic/stochastic $p(y_k|x^{(n)})$. If $y_k$ is continuous, we can quantize $p(y_k|x^{(n)})$. If $p(y_k|x^{(n)})$ is deterministic (i.e., a Dirac delta function), we simply set it to 1 for the value of $y_k$ corresponding to $x^{(n)}$ and 0 for other values of $y_k$. Our metric can also use $p(y_k|x^{(n)})$ from an external expert model. Third, for any particular value $y_k$, we compute $q(z_i|x^{(n)})$ for all $x^{(n)} \in \mathcal{D}$ rather than just for $x^{(n)} \in \mathcal{D}_{y_k}$, which gives more accurate results.

**JEMMIG** A high RMIG value of $y_k$ means that there is a representation $z_{i^*}$ that captures the factor $y_k$. However, $z_{i^*}$ may also capture other factors $y_{\neq k}$ of the data. To make sure that $z_{i^*}$ fits exactly to $y_k$, we provide another metric for interpretability named JEMMIG (standing for Joint Entropy Minuses Mutual Information Gap), computed as follows:

$$\text{JEMMIG}(y_k) \quad = \quad H(z_{i^*}, y_k) - I(z_{i^*}, y_k) + I(z_{j^\circ}, y_k)$$

where $I(z_{i^*}, y_k)$ and $I(z_{j^\circ}, y_k)$ are defined in Eq. 10.

If we estimate $H(z_{i^*}, y_k)$ via quantization, we can bound JEMMIG$(y_k)$ between 0 and $H(y_k) + \log(\#\text{bins})$ (please check Appdx. A.12 for details). A small JEMMIG$(y_k)$ score means that $z_{i^*}$ should match exactly to $y_k$ and $z_{j^\circ}$ should not be related to $y_k$. Thus, we can use JEMMIG$(y_k)$ to *validate whether a model can learn disentangled representations w.r.t a ground truth factor $y_k$ or not* which satisfies the definition in Section 2.1. Note that if we replace $H(z_{i^*}, y_k)$ by $H(y_k)$ to account for the generalization of $z_{i^*}$ over $y_k$, we obtain a metric equivalent to RMIG (but in reverse order).

To compute RMIG and JEMMIG for the whole model, we simply take the average of RMIG$(y_k)$ and JEMMIG$(y_k)$ over all $y_k$ ($k = 1, ..., K$) as follows:

$$\text{RMIG} = \frac{1}{K} \sum_{k=0}^{K-1} \text{RMIG}(y_k) \quad \text{and} \quad \text{JEMMIG} = \frac{1}{K} \sum_{k=0}^{K-1} \text{JEMMIG}(y_k)$$

## 3.4 COMPARISON WITH EXISTING METRICS

In Table 1, we compare our proposed metrics with existing metrics for learning disentangled representations. For deeper analysis of these metrics, we refer readers to Appdx. A.8. One can easily see that only our metrics satisfy the aforementioned robustness criteria. Most other metrics (except for MIG and Modularity) use classifiers, which can cause inconsistent results once the settings of the classifiers change. Moreover, most other metrics (except for MIG) use $\mathbb{E}_{q(z_i|x)}[z_i]$ instead of $q(z_i|x)$ for computing mutual information. This can lead to inaccurate evaluation results since $\mathbb{E}_{q(z_i|x)}[z_i]$ is *theoretically different* from $z_i \sim q(z_i|x)$. Among all metrics, JEMMIG is the only one that can quantify "disentangled representations" defined in Section. 2.1 on its own.

| Metrics | #classifiers | classifier | nonlinear relationship | use $q(z_i|x)$ | continuous factors | real data |
|---|---|---|---|---|---|---|
| Z-diff | 1 | linear/majority-vote | × | × | × | × |
| SAP | $L \times K$ | threshold value | × | × | ✓ | ✓ |
| MIG | 0 | none | ✓ | ✓ | × | × |
| Disentanglement | | | | | | |
| Completeness | $K$ | LASSO/ random forest | ×/✓ | × | × | × |
| Informativeness | | | | | | |
| Modularity | 0 | none | ✓ | × | ✓ | × |
| Explicitness | $K$ | one-vs-rest logistic regressor | × | × | × | × |
| WSEPIN† | 0 | none | ✓ | ✓ | ✓ | ✓ |
| WINDIN† | 0 | none | ✓ | ✓ | ✓ | ✓ |
| RMIG | 0 | none | ✓ | ✓ | ✓ | ✓ |
| JEMMIG* | 0 | none | ✓ | ✓ | ✓ | ✓ |

Table 1: Analysis of different metrics for disentanglement learning. $L$ and $K$ are the numbers of latent variables and ground truth factors, respectively. Metrics marked with * are self-contained. Metrics marked with † do not require ground truth factors of variation.

## 4 EXPERIMENTS

We use our proposed metrics to evaluate three representation learning methods namely FactorVAE (Kim & Mnih, 2018), $\beta$-VAE (Higgins et al., 2017a) and AAE (Makhzani et al., 2015) on *both real and toy datasets* which are CelebA (Liu et al., 2015) and dSprites (Matthey et al., 2017), respectively. A brief discussion of these models are given in Appdx. A.1. We would like to show the following points: i) how to compare models based on our metrics; ii) the advantages of our metrics compared to other metrics; iii) the consistence between qualitative results produced by our metrics and visual results; and iv) the ablation study of our metrics.

Due to space limit, we only present experiments for the first two points. The experiments for points (iii) and (iv) are put in Appdx. A.4 and Appdx. A.5, respectively. Details about the datasets and model settings are provided in Appdx. A.2 and Appdx. A.3, respectively. In all figures below, "TC" refers to the $\gamma$ coefficient of the TC loss in FactorVAEs (Kim & Mnih, 2018), "Beta" refers to the $\beta$ coefficient in $\beta$-VAEs (Higgins et al., 2017a).

**Informativeness** In Figs. 2a and 2b, we show the average amount of information (of $x$) that a representation $z_i$ contains (the mean of $I(z_i, x)$) and the total amount of information that all representations $z$ contain ($I(z, x)$). It is clear that adding the TC term to the standard VAE loss does not affect $I(z, x)$ much (Fig. 2b). However, because $z_i$ and $z_j$ in FactorVAEs are more separable than those in standard VAEs, FactorVAEs should produce smaller $I(z_i, x)$ than standard VAEs on average (Fig. 2a). We also see that the mean of $I(z_i, x)$ and $I(z, x)$ consistently decrease for $\beta$-VAEs with higher $\beta$.

**Separability and Independence** If we only evaluate models based on the separability of representations, $\beta$-VAE models with large $\beta$ are among the best. These models force latent representations

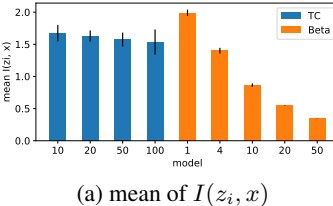

(a) mean of $I(z_i, x)$

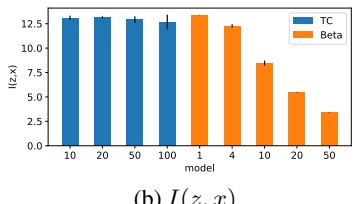

(b) $I(z, x)$

Figure 2: The informativeness and the total information of some FactorVAE and $\beta$-VAE models. For each hyperparameter, we report the mean and the standard error of 4 different runs.

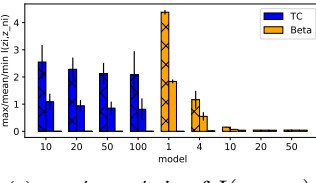

(a) max/mean/min of $I(z_i, z_{\neq i})$

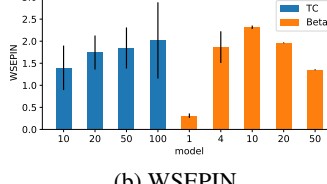

(b) WSEPIN

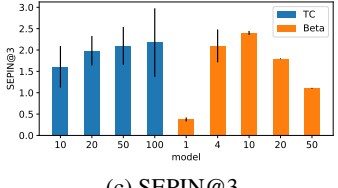

(c) SEPIN@3

Figure 3: $I(z_i, z_{\neq i})$, WSEPIN and SEPIN@3 of some FactorVAE and $\beta$-VAE models.

to be highly separable (as in Fig. 3a, we can see that the max/mean/min values of $I(z_i, z_{\neq i})$ are equally small for $\beta$-VAEs with large $\beta$). In FactorVAEs, informative representations usually have poor separability (large value) and noisy representations usually have perfect separability ($\approx 0$) (Fig. 4a). Increasing the weight of the TC loss improves the max and mean of $I(z_i, z_{\neq i})$ but not significance (Fig. 3a).

Using WSEPIN and SEPIN@3 gives us a more reasonable evaluation of the disentanglement capability of these models. In Fig. 3b, we see that $\beta$-VAE models with $\beta = 10$ achieve the highest WSEPIN and SEPIN@3 scores, which suggests that their informative representations usually contain large amount of information of $x$ that are *not shared by other representations*. However, this type of information may not associate well with the ground truth factors of variation (e.g., $z_3, z_6$ in Fig. 4c). The representations of FactorVAEs, despite containing less information of $x$ *on their own*, usually reflect the ground truth factors more accurately (e.g., $z_5, z_8, z_7$ in Fig. 4a) than those of $\beta$-VAEs. These results suggest that ground truth factors should be used for proper evaluations of disentanglement.

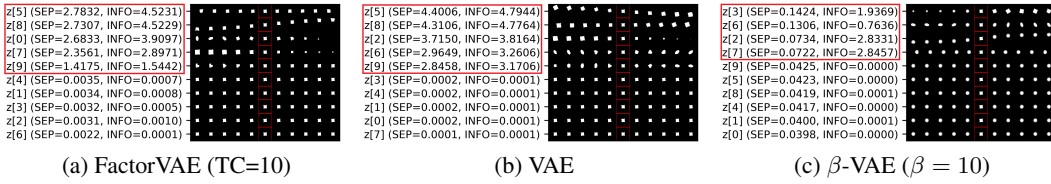

(a) FactorVAE (TC=10)        (b) VAE        (c) $\beta$-VAE ($\beta = 10$)

Figure 4: Visualization of the representations learned by representative FactorVAE, VAE, and $\beta$-VAE models with separability ($I(z_i, z_{\neq i})$) and informativeness ($I(z_i, x)$) scores. Representations are sorted by their separability scores.

**Interpretability**    Using JEMMIG and RMIG, we see that FactorVAE models can learn representations that are more interpretable than those learned by $\beta$-VAE models. Surprisingly, the worst FactorVAE models (with TC=10) clearly outperform the best $\beta$-VAE models (with $\beta = 10$). This result is sensible because it is accordant with the visualization in Figs. 4a and 4c.

**Comparison with Z-diff**    In (Chen et al., 2018), the authors have already shown that MIG is more robust than Z-diff (Higgins et al., 2017a) so we compare our metrics with MIG directly.

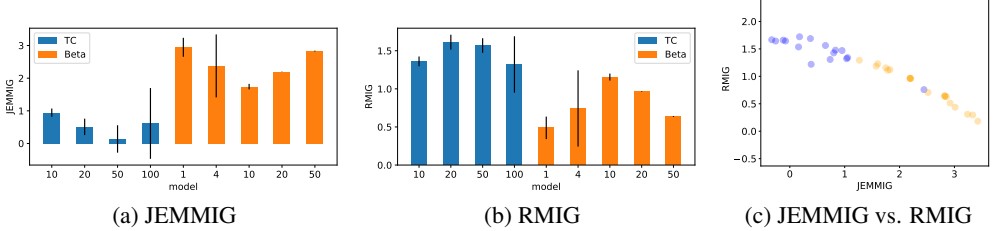

Figure 5: (a) and (b): Unnormalized JEMMIG and RMIG scores of several FactorVAE and $\beta$-VAE models. (c): Correlation between JEMMIG and RMIG.

**Comparison with MIG** On toy datasets like dSprites, RMIG produces similar results as MIG (Chen et al., 2018). Please check Appdx. A.13 for more details.

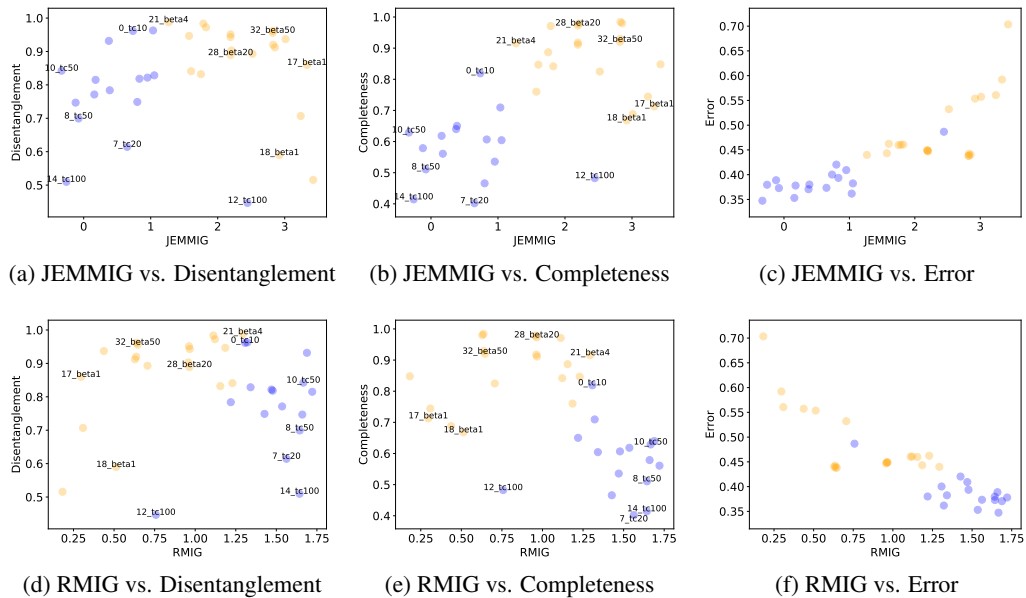

Figure 6: Comparison between JEMMIG/RMIG and the metrics in (Eastwood & Williams, 2018). Because the competing metrics do not apply for categorical factors (see Appdx. A.8 for detailed analysis), we exclude the "shape" factor during computation. Following (Eastwood & Williams, 2018), we use LASSO classifiers with the L1 coefficient is $\alpha = 0.002$. Blue dots denote FactorVAE models and orange dots denote $\beta$-VAE models.

**Comparison with "disentanglement", "completeness" and "informativeness"** In Fig. 6, we show the differences in evaluation results between JEMMIG/RMIG and the metrics in (Eastwood & Williams, 2018). We can easily see that JEMMIG and RMIG are much better than "disentanglement", "completeness" and "informativeness" (or reversed classification error) in separating FactorVAE and $\beta$-VAE models. Among the three competing metrics, only "informativeness" (or $I(z, y_k)$) seems to be correlated with JEMMIG and RMIG. This is understandable because when most representations are independent in case of FactorVAEs and $\beta$-VAEs, we have $I(z, y_k) \approx I(z_{i^*}, y_k) \approx I(z_{i^*}, y_k) - I(z_{j^\circ}, y_k)$. "Disentanglement" and "completeness", by contrast, are strongly uncorrelated with JEMMIG and RMIG. While JEMMIG consistently grades standard VAEs ($\beta = 1$) worse than other models (Fig. 5a), "disentanglement" and "completeness" usually grade standard VAEs better than some FactorVAE models, which seems inappropriate. Moreover, since "disentanglement" and "completeness" are not well aligned, using both of them at the same time may cause confusion. For example, the model "28_beta20" has lower "disentanglement" score yet higher "completeness" score

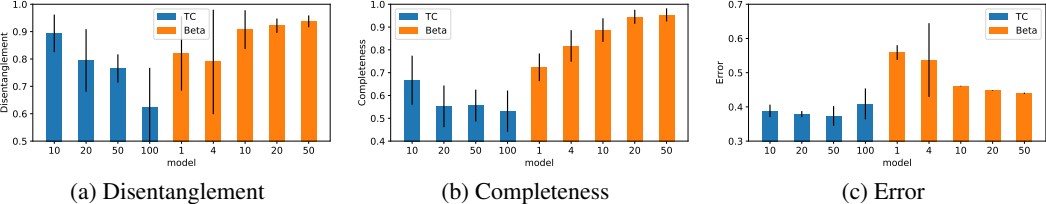

| (a) Disentanglement | (b) Completeness | (c) Error |

Figure 7: "disentanglement", "completeness" and "informativeness" (error) scores of several Factor-VAE and $\beta$-VAE models.

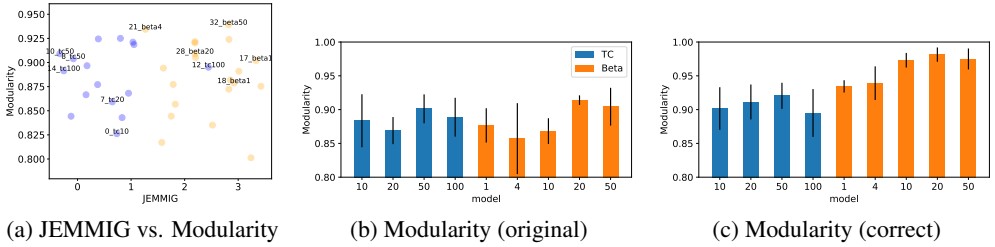

| (a) JEMMIG vs. Modularity | (b) Modularity (original) | (c) Modularity (correct) |

Figure 8: (a): Comparison between JEMMIG and "modularity" (#bins=100). (b) and (c): "modularity" scores of several FactorVAE and $\beta$-VAE models. The original version computes $I(z_i, y_k)$ using $\mathbb{E}_{q(z_i|x)}[z_i]$ while the correct version compute $I(z_i, y_k)$ using $q(z_i|x)$.

than the model "32_beta50" (Figs. 6a and 6b) so it is hard to know which model is better than the other at learning disentangled representations.

From Figs. 7a and 7b, we see that "disentanglement" and "completeness" blindly favor $\beta$-VAE models with high $\beta$ without concerning about the fact that representations in these models are less informative than representations in FactorVAEs (Fig. 7c). Thus, they are not good for characterizing disentanglement in general.

Disentanglement and "completeness" are computed based on a weight matrix with an assumption that the weight magnitudes for noisy representations are close to 0. However, this assumption is often broken in practice, thus, may lead to inaccurate results (please check Appdx. A.9 for details).

**Comparison with "modularity"** "modularity" and "explicitness" (Ridgeway & Mozer, 2018) are similar to "disentanglement" and "informativeness" (Eastwood & Williams, 2018) in terms of concept, respectively. However, they are different in terms of formulation. We exclude "explicitness" in our experiment because computing it on dSprites is time consuming. In Fig. 8a, we show the correlation between JEMMIG and "modularity". We consider two versions of "modularity". In the first version (Fig. 8b), $I(z_i, y_k)$ is computed from the mean of $z_i \sim q(z_i|x)$. This is the original implementation provided by (Ridgeway & Mozer, 2018). In the second version (Fig. 8c), $I(z_i, y_k)$ is computed from $q(z_i|x)$. We can see that in either case, "modularity" often gives higher scores for standard VAEs than for FactorVAEs. It means that like "disentanglement", "modularity" itself does not fully specify "disentangled representations" defined in Section 2.1.

## 5 CONCLUSION

We have proposed an information-theoretic definition of disentangled representations and designed robust metrics for evaluation, along three dimensions: *informativeness*, *separability* and *interpretability*. We carefully analyze the properties of our metrics using well known representation learning models namely FactorVAE, $\beta$-VAE and AAE on both real and toy datasets. Compared with existing metrics, our metrics are more robust and produce more sensible evaluations that are compatible with visual results. Based on our definition of disentangled representation in Section 2.1, WSEPIN/JEMMIG are the two key metrics in case ground truth labels are unavailable/available, respectively.

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

# A APPENDIX

## A.1 REVIEW OF FACTORVAES, $\beta$-VAES AND AAES

Standard VAEs are trained by minimizing the variational upper bound $\mathcal{L}^{\text{VAE}}$ of $-\log p_\theta(x)$ as follows:

$$\mathcal{L}^{\text{VAE}} = \mathbb{E}_{p_\mathcal{D}(x)} \left[ \mathbb{E}_{q_\phi(z|x)} \left[ -\log p_\theta(x|z) \right] + D_{KL} \left( q_\phi(z|x) \| p(z) \right) \right] \qquad (11)$$

where $q_\phi(z|x)$ is an amortized variational posterior distribution. However, this objective does not lead to disentangled representations (Higgins et al., 2017a).

$\beta$-VAEs (Higgins et al., 2017a) penalize the KL term in the original VAE loss more heavily with a coefficient $\beta \gg 1$:

$$\mathcal{L}^{\beta\text{-VAE}} = \mathbb{E}_{p_\mathcal{D}(x)} \left[ \mathbb{E}_{q_\phi(z|x)} \left[ -\log p_\theta(x|z) \right] + \beta D_{KL} \left( q_\phi(z|x) \| p(z) \right) \right]$$

Since $\mathbb{E}_{p_\mathcal{D}(x)} \left[ D_{KL} \left( q_\phi(z|x) \| p(z) \right) \right] = I_\phi(x, z) + D_{KL} \left( q_\phi(z) \| p(z) \right)$, more penalty on the KL term encourages $q_\phi(z)$ to be factorized but also forces $z$ to discard more information in $x$.

FactorVAEs (Kim & Mnih, 2018) add a constraint to the standard VAE loss to explicitly impose factorization of $q_\phi(z)$:

$$\mathcal{L}^{\text{FactorVAE}} = \mathcal{L}^{\text{VAE}} + \gamma D_{KL} \left( q_\phi(z) \| \prod_i q_i(z_i) \right) \qquad (12)$$

where $D_{KL} \left( q_\phi(z) \| \prod_i q_i(z_i) \right) \geq 0$ is known as the *total correlation* (TC) of $z$. Intuitively, $\gamma$ can be large without affecting the mutual information between $z$ and $x$, making FactorVAE more robust than $\beta$-VAE in learning disentangled representations. Other related models that share similar ideas with FactorVAEs are are $\beta$-TCVAEs (Chen et al., 2018) and DIP-VAEs (Kumar et al., 2017).

The loss of AAEs (Makhzani et al., 2015) is derived from the standard VAE loss by removing the term $I_\phi(x, z)$:

$$\mathcal{L}^{\text{AAE}} = \mathbb{E}_{p_\mathcal{D}(x)} \left[ \mathbb{E}_{q_\phi(z|x)} \left[ -\log p_\theta(x|z) \right] \right] + D_{KL} \left( q_\phi(z) \| p(z) \right)$$

Different from the losses of $\beta$-VAEs and FactorVAEs, AAE loss is not a valid upper bound on $-\log p_\theta(x)$.

## A.2 DATASETS

The CelebA dataset (Liu et al., 2015) consists of more than 200 thousands face images with 40 binary attributes. We resize these images to $64 \times 64$. The dSprites dataset (Matthey et al., 2017) is a toy dataset generated from 5 different factors of variation which are "shape" (3 values), "scale" (6 values), "rotation" (40 values), "x-position" (32 values), "y-position" (32 values). Statistics of these datasets are provided in Table 2.

| Dataset | #Train | #Test | Image size |
|---------|--------|-------|------------|
| CelebA | 162,770 | 19,962 | 64×64×3 |
| dSprites | 737,280 | 0 | 64×64×1 |

Table 2: Summary of datasets used in experiments.

## A.3 MODEL SETTINGS

For FactorVAE, $\beta$-VAE and AAE, we used the same architectures for the encoder and decoder (see Table 3 and Table 4[5]), following (Kim & Mnih, 2018). We trained the models for 300 epochs with mini-batches of size 64. The learning rate is $10^{-3}$ for the encoder/decoder and is $10^{-4}$ for the discriminator over $z$. We used Adam (Kingma & Ba, 2014) optimizer with $\beta_1 = 0.5$ and $\beta_2 = 0.99$. Unless explicitly mentioned, we use the following default settings: i) for CelebA: the number of

---

[5]Only FactorVAE and AAE use a discriminator over $z$

latent variables is 65, the TC coefficient in FactorVAE is 50, the value for $\beta$ in $\beta$-VAE is 50, and the coefficient for the generator loss over $z$ in AAE is 50; ii) for dSprites: the number of latent variables is 10.

| Encoder | Decoder | Discriminator Z |
|---|---|---|
| $x$ dims: 64×64×3 | $z$ dim: 65 | $z$ dim: 65 |
| conv (4, 4, 32), stride 2, ReLU | FC 1×1×256, ReLU | 5×[FC 1000, LReLU] |
| conv (4, 4, 32), stride 2, ReLU | deconv (4, 4, 64), stride 1, valid, ReLU | FC 1 |
| conv (4, 4, 64), stride 2, ReLU | deconv (4, 4, 64), stride 2, ReLU | $D(z)$: 1 |
| conv (4, 4, 64), stride 2, ReLU | deconv (4, 4, 32), stride 2, ReLU | |
| conv (4, 4, 256), stride 1, valid, ReLU | deconv (4, 4, 32), stride 2, ReLU | |
| FC 65 | deconv (4, 4, 3), stride 2, ReLU | |
| $z$ dim: 65 | $x$ dim: 64×64×3 | |

Table 3: Model architectures for CelebA.

| Encoder | Decoder | Discriminator Z |
|---|---|---|
| $x$ dims: 64×64×1 | $z$ dim: 10 | $z$ dim: 10 |
| conv (4, 4, 32), stride 2, ReLU | FC 128, ReLU | 5×[FC 1000, LReLU] |
| conv (4, 4, 32), stride 2, ReLU | FC 4×4×64, ReLU | FC 1 |
| conv (4, 4, 64), stride 2, ReLU | deconv (4, 4, 64), stride 2, ReLU | $D(z)$: 1 |
| conv (4, 4, 64), stride 2, ReLU | deconv (4, 4, 32), stride 2, ReLU | |
| FC 128, ReLU | deconv (4, 4, 32), stride 2, ReLU | |
| FC 10 | deconv (4, 4, 1), stride 2, ReLU | |
| $z$ dim: 10 | $x$ dim: 64×64×1 | |

Table 4: Model architecture for dSprites.

## A.4 CONSISTENCE BETWEEN QUANTITATIVE AND QUALITATIVE RESULTS

### A.4.1 CELEBA

**Informativeness** We sorted the representations of different models according to their informativeness scores in the descending order and plot the results in Fig. 9. There are distinct patterns for different methods. AAE captures equally large amounts of information from the data while FactorVAE and $\beta$-VAE capture smaller and varying amounts. This is because FactorVAE and $\beta$-VAE penalize the informativeness of representations while AAE does not. Recall that $I(z_i, x) = H(z_i) - H(z_i|x)$. For AAE, $H(z_i|x) = 0$ and $H(z_i)$ is equal to the entropy of $\mathcal{N}(0, \mathrm{I})$. For FactorVAE and $\beta$-VAE, $H(z_i|x) > 0$ and $H(z_i)$ is usually smaller than the entropy of $\mathcal{N}(0, \mathrm{I})$ due to a narrow $q(z_i)$[6].

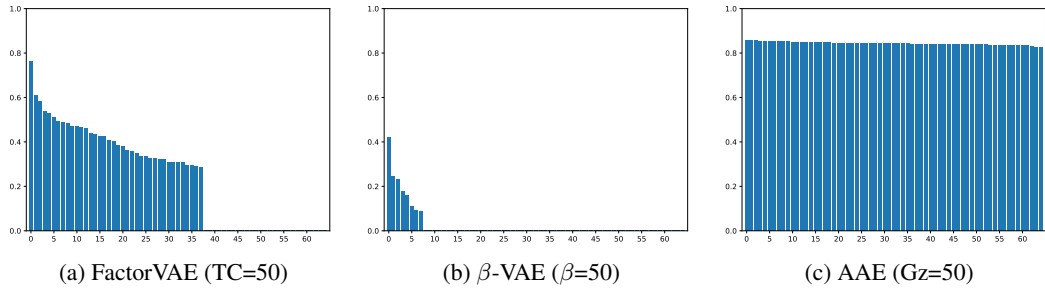

(a) FactorVAE (TC=50)  (b) $\beta$-VAE ($\beta$=50)  (c) AAE (Gz=50)

Figure 9: Normalized informativeness scores (bins=100) of all latent variables sorted in descending order.

---

[6]Note that $H(z_i)$ does not depend on whether $q(z_i)$ is zero-centered or not

In Fig. 9, we see a sudden drop of the scores to 0 for some FactorVAE's and $\beta$-VAE's representations. These representations $z_i$ are totally random and contain no information about the data (i.e., $q(z_i|x) \approx \mathcal{N}(0, \mathrm{I})$). We call them "noisy" representations and provide discussions in Appdx. A.7.

We visualize the top 10 most informative representations for these models in Fig. 10. AAE's representations are more detailed than FactorVAE's and $\beta$-VAE's, suggesting the effect of high informativeness. However, AAE's representations mainly capture information within the support of $p_{\mathcal{D}}(x)$. This explains why we still see a face when interpolating AAE's representations. By contrast, FactorVAE's and $\beta$-VAE's representations usually contain information outside the support of $p_{\mathcal{D}}(x)$. Thus, when we interpolate these representations, we may see something not resembling a face.

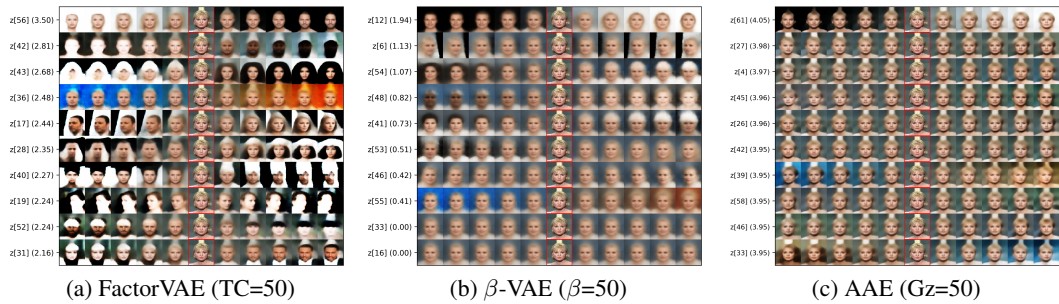

(a) FactorVAE (TC=50)      (b) $\beta$-VAE ($\beta$=50)      (c) AAE (Gz=50)

Figure 10: Visualization of the top informative representations. Scores are unnormalized.

**Separability and Independence** Table 5 reports MISJED scores (Section 3.2) for the top most informative representations. FactorVAE achieves the lowest MISJED scores, AAE comes next and $\beta$-VAE is the worst. We argue that this is because FactorVAE learns independent and nearly deterministic representations, $\beta$-VAE learns strongly independent yet highly stochastic representations, and AAE, on the other extreme side, learns strongly deterministic yet not very independent representations. From Table 5 and Fig. 11, it is clear that MISJED produces correct orders among pairs of representations according to their informativeness.

| | MISJED (unnormalized) | | | | | |
| --- | --- | --- | --- | --- | --- | --- |
| | $z_1, z_2$ | $z_1, z_3$ | $z_1, z_{-1}$ | $z_1, z_{-2}$ | $z_{-1}, z_{-2}$ | $z_{-1}, z_{-3}$ |
| FactorVAE | **0.008** | **0.009** | 2.476 | 2.443 | 4.858 | 4.892 |
| $\beta$-VAE | 0.113 | 0.131 | 3.413 | 3.401 | 6.661 | 6.739 |
| AAE | 0.022 | 0.023 | 0.022 | 0.021 | 0.021 | 0.020 |

Table 5: Unnormalized MISJED scores (#bins = 50, 10% data). $z_1, z_2, z_3$ and $z_{-1}, z_{-2}, z_{-3}$ denote the top 3 and the bottom 3 latent variables sorted by the informativeness scores in descending order. Boldness indicates best results.

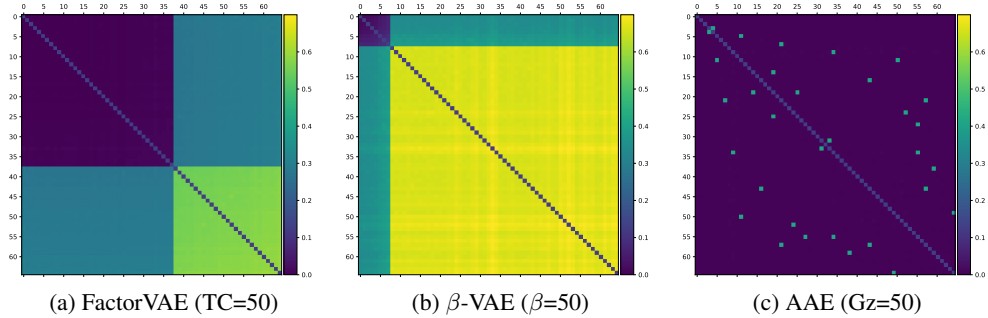

(a) FactorVAE (TC=50)      (b) $\beta$-VAE ($\beta$=50)      (c) AAE (Gz=50)

Figure 11: Normalized MISJED scores of all latent pairs sorted by their informativeness.

**Interpretability**   We report the RMIG scores and JEMMIG scores for several ground truth factors in the CelebA dataset in Tables 6 and 7, respectively. In general, FactorVAE learns representations that agree better with the ground truth factors than $\beta$-VAE and AAE do. This is consistent with the qualitative results in Fig. 12. However, all models still perform poorly for interpretability since their RMIG and JEMMIG scores are very far from 1 and 0, respectively. We provide the normalized JEMMIG and RMIG scores for all attributes in Fig. 13.

| | RMIG (normalized) | | | | | |
|---|---|---|---|---|---|---|
| | Bangs | Black Hair | Eyeglasses | Goatee | Male | Smiling |
| | H=0.4256 | H=0.5500 | H=0.2395 | H=0.2365 | H=0.6801 | H=0.6923 |
| FactorVAE | **0.1742** | **0.0430** | **0.0409** | **0.0343** | 0.0060 | **0.0962** |
| $\beta$-VAE | 0.0176 | 0.0223 | 0.0045 | 0.0325 | **0.0094** | 0.0184 |
| AAE | 0.0035 | 0.0276 | 0.0018 | 0.0069 | 0.0060 | 0.0099 |

Table 6: Normalized RMIG scores (#bins=100) for some factors. Higher is better.

| | JEMMIG (normalized) | | | | | |
|---|---|---|---|---|---|---|
| | Bangs | Black Hair | Eyeglasses | Goatee | Male | Smiling |
| | H=0.4256 | H=0.5500 | H=0.2395 | H=0.2365 | H=0.6801 | H=0.6923 |
| FactorVAE | **0.6118** | **0.6334** | **0.6041** | **0.6616** | **0.6875** | **0.6150** |
| $\beta$-VAE | 0.8632 | 0.8620 | 0.8602 | 0.8600 | 0.8690 | 0.8699 |
| AAE | 0.8463 | 0.8613 | 0.8423 | 0.8496 | 0.8644 | 0.8575 |

Table 7: Normalized JEMMIG scores (#bins=100) for some factors. Lower is better.

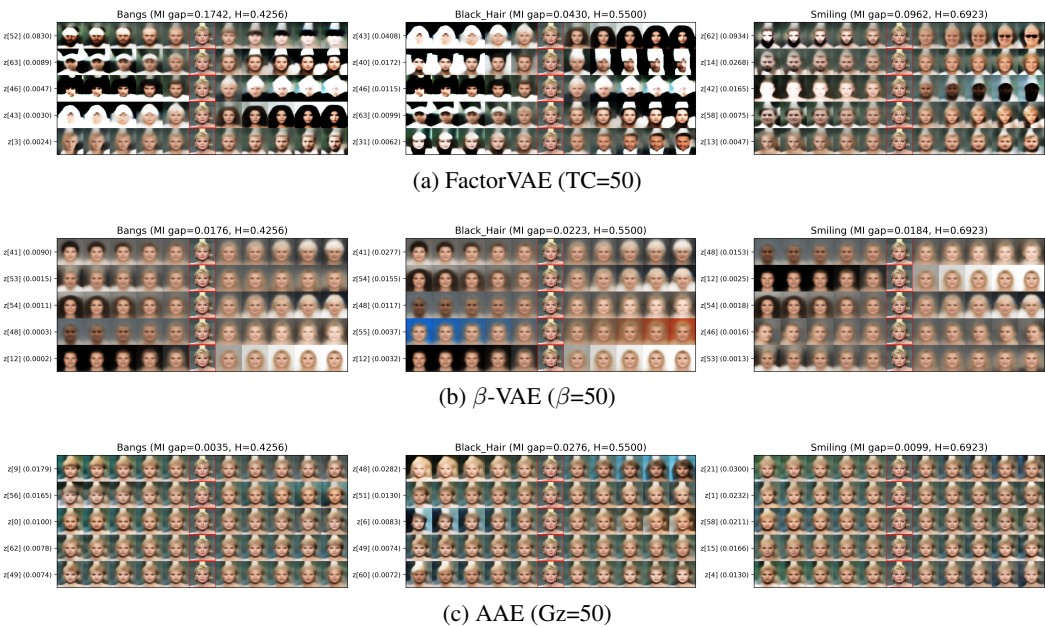

(a) FactorVAE (TC=50)

(b) $\beta$-VAE ($\beta$=50)

(c) AAE (Gz=50)

Figure 12: Top 5 representations that are most correlated with some ground truth factors. For each representation, we show its mutual information with the ground truth factor.

### A.4.2   DSPRITES

**Informativeness**   From Fig. 14, we see that 5 representations of AAE have equally high informativeness scores while the remaining 5 representations have nearly zeros informativeness scores. This is because AAE needs only 5 representations to capture all information in the data. FactorVAE also

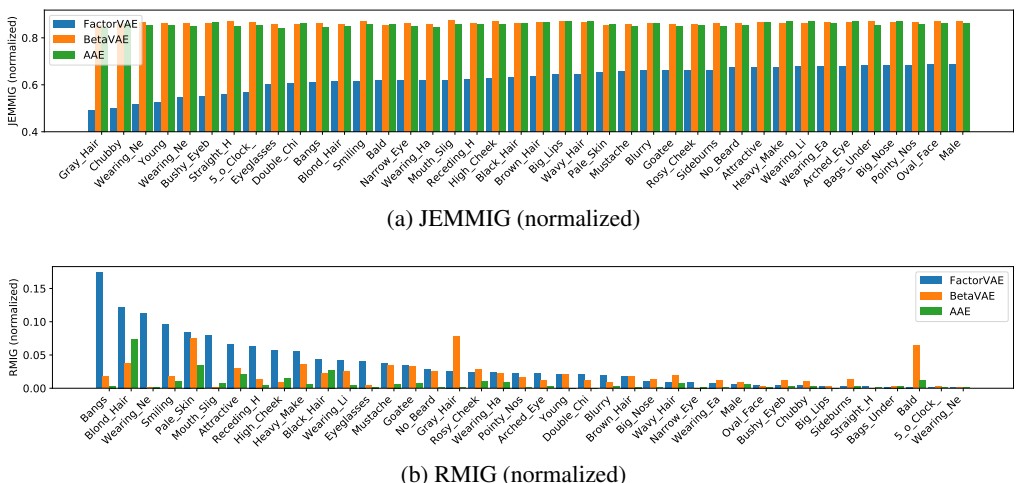

(a) JEMMIG (normalized)

(b) RMIG (normalized)

Figure 13: Normalized JEMMIG and RMIG scores for all attributes in the CelebA dataset. We sorted the JEMMIG and RMIG scores of the FactorVAE in ascending and descending orders, respectively.

needs only 5 representations but some are less informative than those of AAE. Note that the number of ground truth factors of variation in dSprites dataset is also 5.

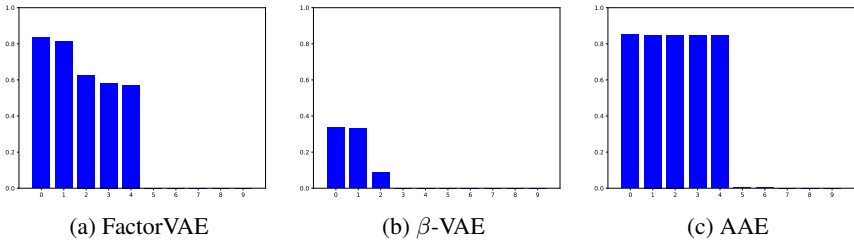

(a) FactorVAE        (b) $\beta$-VAE        (c) AAE

Figure 14: Normalized informativeness scores (bins=100) of all latent variables sorted in descending order.

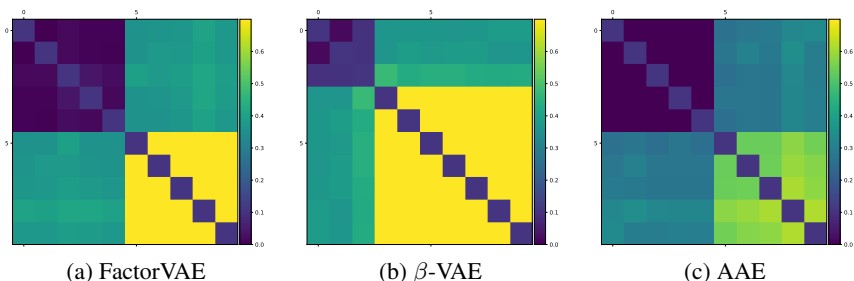

(a) FactorVAE        (b) $\beta$-VAE        (c) AAE

Figure 15: Normalized MISJED scores (bins=100) of all latent pairs sorted by their informativeness.

**Separability and Independence**    Fig. 15 shows heat maps of MISJED scores for the three models.

**Interpretability**    From Tables. 8 and 9, we see that FactorVAE is very good at disentangling "scale", "x-position" and "y-position" but fails to disentangling "shape" and "rotation". However, FactorVAE still performs much better than $\beta$-VAE and AAE. These results are consistent with the visual results in Fig. 16.

Also note that in FactorVAE, the RMIG scores for "scale" and "x-position" are quite similar but the JEMMIG score for "scale" is higher than that for "x-position". This is because the quantized distribution (with 100 bins) of a particular representation $z_i$ fits better to the distribution of "x-position" (having 32 possible values) than to the distribution of "scale" (having only 6 possible values).

|  | Shape | Scale | Rotation | Pos X | Pos Y |
|---|---|---|---|---|---|
| FactorVAE | **0.2412** | **0.7139** | **0.0523** | **0.7198** | **0.7256** |
| $\beta$-VAE | 0.0481 | 0.1533 | 0.0000 | 0.4127 | 0.4193 |
| AAE | 0.0053 | 0.0786 | 0.0098 | 0.3932 | 0.4509 |

Table 8: Normalized RMIG scores (bins=100).

|  | Shape | Scale | Rotation | Pos X | Pos Y |
|---|---|---|---|---|---|
| FactorVAE | **0.6841** | **0.3422** | **0.7204** | **0.2908** | **0.2727** |
| $\beta$-VAE | 0.8642 | 0.8087 | 0.9199 | 0.5629 | 0.5576 |
| AAE | 0.8426 | 0.8143 | 0.8665 | 0.5738 | 0.5258 |

Table 9: Normalized JEMMIG scores (bins=100).

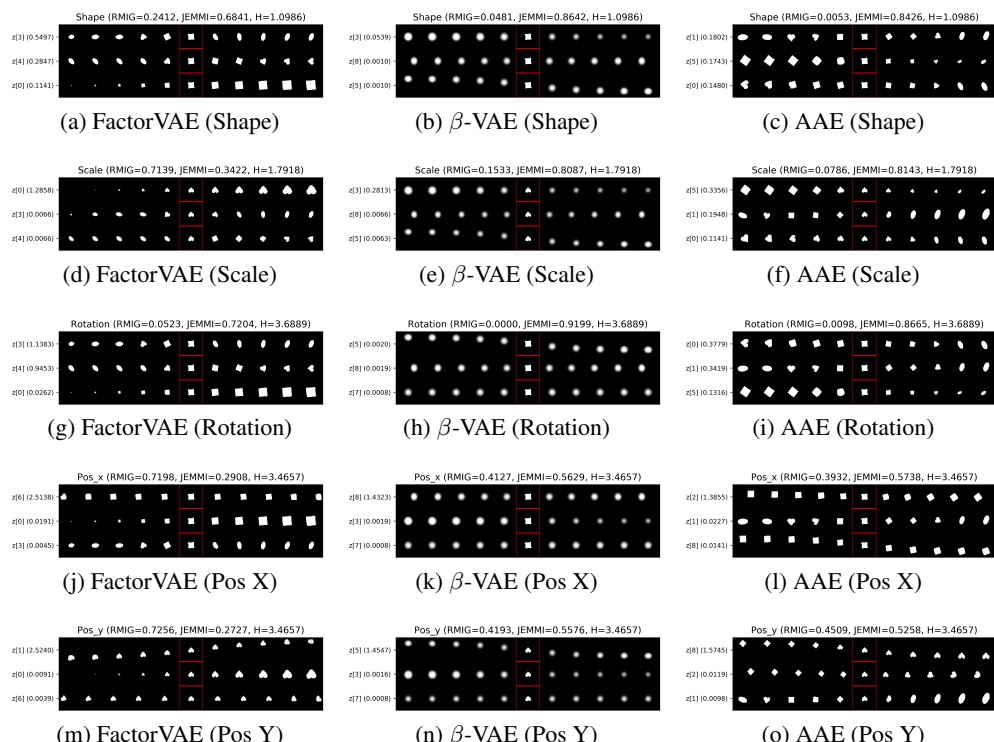

(a) FactorVAE (Shape)   (b) $\beta$-VAE (Shape)   (c) AAE (Shape)

(d) FactorVAE (Scale)   (e) $\beta$-VAE (Scale)   (f) AAE (Scale)

(g) FactorVAE (Rotation)   (h) $\beta$-VAE (Rotation)   (i) AAE (Rotation)

(j) FactorVAE (Pos X)   (k) $\beta$-VAE (Pos X)   (l) AAE (Pos X)

(m) FactorVAE (Pos Y)   (n) $\beta$-VAE (Pos Y)   (o) AAE (Pos Y)

Figure 16: Top 3 representations sorted by their mutual information with different ground truth factors.

## A.5 ABLATION STUDY OF OUR METRICS

**Sensitivity of the number of bins** When estimating entropy and mutual information terms using quantization, we need to specify the value range and the number of bins (#bins) in advance. In this paper, we fix the value range to be [-4, 4] since most latent values fall within this range. We only investigate the effect of #bins on the RMIG and JEMMIG scores for different models and show the results in Fig. 17 (left, middle).

We can see that when #bins is small, RMIG scores are low. This is because the quantized distributions $Q(z_{i^*})$ and $Q(z_{j^\circ})$ look similar, causing $I^*(z_{i^*}, y_k)$ and $I^\circ(z_{j^\circ}, y_k)$ to be similar as well. When #bins is large, the quantized distribution $Q(z_{i^*})$ and $Q(z_{j^\circ})$ look more different, leading to higher RMIG scores. RMIG scores are stable when #bins > 200, which suggests that finer quantizations do not affect the estimation of $I(z_i, y_k)$ much.

Unlike RMIG scores, JEMMIG scores keep increasing when we increase #bins. Note that JEMMIG only differs from RMIG in the appearance of $H(z_{i^*}, y_k)$. Finer quantizations of $z_{i^*}$ introduce more information about $z_{i^*}$, hence, always lead to higher $H(z_{i^*}, y_k)$ (see Fig. 17 (right)). Larger JEMMIG scores also reflect the fact that finer quantizations of $z_{i^*}$ make $z_{i^*}$ look more *continuous*, thus, less interpretable w.r.t the *discrete* factor $y_k$.

We provide a detailed explanation about the behaviors of RMIG and JEMMIG w.r.t #bins in Appdx. A.11. Despite the fact that #bins affects the RMIG and JEMMIG scores of a single model, the relative order among different models remains the same. It suggests that once we fixed the #bins, we can use RMIG and JEMMIG scores to rank different models.

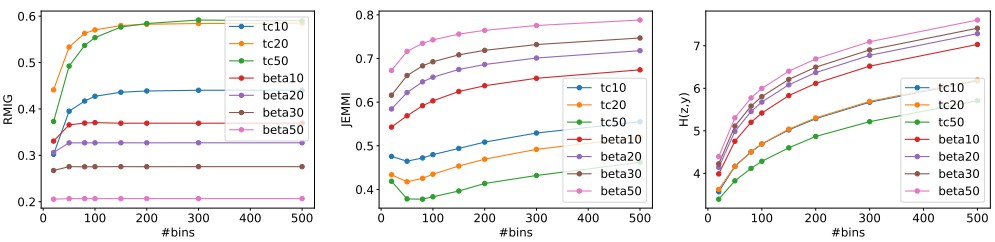

Figure 17: Dependences of RMIG (normalized), JEMMIG (normalized) and $\frac{1}{K}\sum_{k=0}^{K-1} H(z_{i^*}, y_k)$ on the number of bins. The dataset is dSprite.

**Sensitivity of the number of samples** From Fig. 18 (left, right), it is clear that the sampling estimation is unbiased and is not affected much by the number of samples.

**Sensitivity of sampling in high dimensional space** One thing that we should concern about is the performance of our metrics when the number of latent representations (#latents) is large (or $z$ is high-dimensional). In Fig. 19a, we see that the informativeness of an individual representations $z_i$ is not affected by #latents. When we increase #latents, additional representations are usually noisy ($I(z_i, x) \approx 0$). The total amount of information captured by the model ($I(x, z)$), by contrast, highly depends on #latents (Fig. 19b). Unusually, increasing #latents *reduces* $I(x, z)$ instead of increasing it. We have not found the final answer for this phenomenon but possible hypotheses are: i) On a high dimensional space where most latent representations are noisy (e.g. #latents=20), $q(z|x)$ may look more similar to $q(z)$, causing the wrong calculation of $\log\frac{q(z|x)}{q(z)}$, or ii) when #latents is large, $q(z|x) = \prod_{i=0}^{L-1} q(z_i|x)$ is very tiny, thus, may lead to floating point imprecision[7]. In Fig. 19c, we

---

[7]We tried $q(z|x) = \exp\left(\sum_{i=0}^{L-1} \log q(z_i|x)\right)$ and it gives similar results as $q(z|x) = \prod_{i=0}^{L-1} q(z_i|x)$.

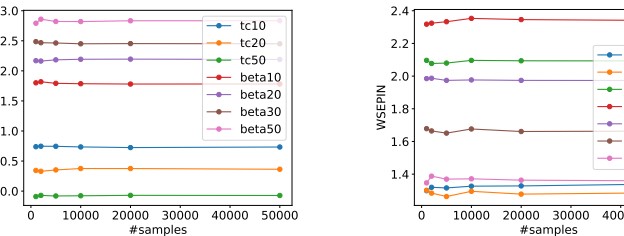

Figure 18: Dependences of JEMMIG and WSEPIN on the number of samples. All models have 10 latent variables. The dataset is dSprites.

see that increasing #latents increases $I(z_i, z_{\neq i})$. This makes sense because larger #latents means that $z_{\neq i}$ will contain more information. However, the change of $I(z_i, z_{\neq i})$ is sudden when #latents change from 10 to 15, which is different from the change of #latents from 5 to 10 or 15 to 20. Recall that $I(z_i, z_{\neq i}) = H(z_i) + H(z_{\neq i}) - H(z)$. Since $H(z_i)$ can be computed stably, we only plot $H(z_{\neq i}) - H(z)$ and show it in Fig. 19d. We can see that when #latents = 20, $H(z_{\neq i}) \approx H(z)$ which means we cannot differentiate between $q(z_{\neq i})$ and $q(z)$. The instability of computation for high dimensional latents becomes clearer in Fig. 19e as $I(x, z_i | z_{\neq i}) = I(x, z) - I(x, z_{\neq i})$ can be $< 0$ when #latents = 15 or 20. This causes the instability of WSEPIN in Fig. 19f despite the results look reasonable. JEMMIG and RMIG are calculated on individual latents so they are not affected by #latents and can provide consistent evaluations for models with different #latents.

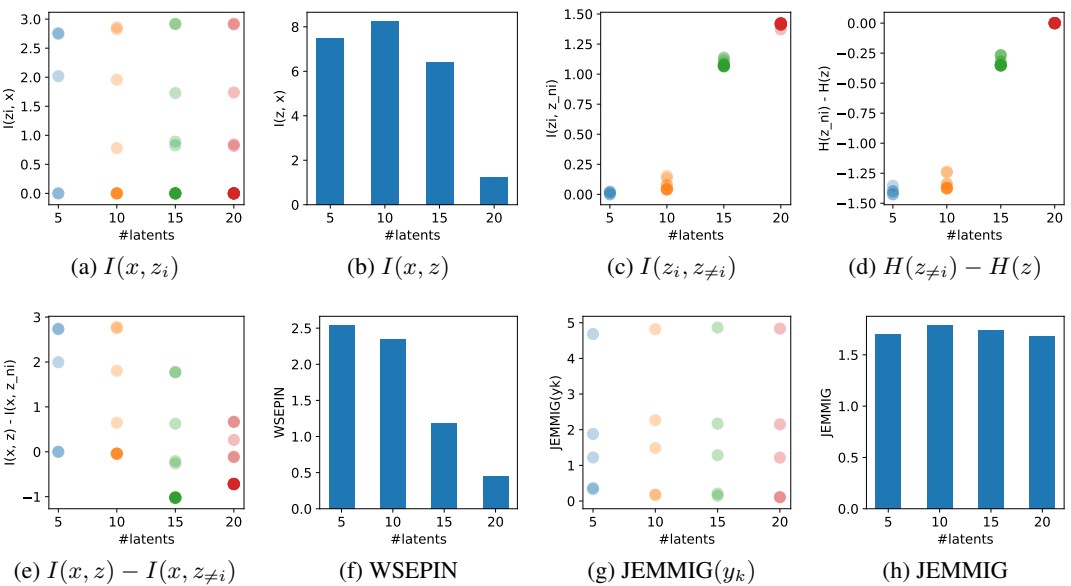

(a) $I(x, z_i)$      (b) $I(x, z)$      (c) $I(z_i, z_{\neq i})$      (d) $H(z_{\neq i}) - H(z)$

(e) $I(x, z) - I(x, z_{\neq i})$      (f) WSEPIN      (g) JEMMIG($y_k$)      (h) JEMMIG

Figure 19: Dependences of various quantitative measures on the number of latents. All measures are computed via sampling. The model used in this experiment is $\beta$-VAE with $\beta = 10$.

## A.6    EVALUATING INDEPENDENCE WITH CORRELATION MATRIX

For every $x^{(n)}$ sampled from the training data, we generated $m = 1$ latent samples $z_i^{(n,m)} \sim q(z_i | x^{(n)})$ and built a correlation matrix from these samples for each of the models FactorVAE, $\beta$-VAE and AAE. We also built another version of the correlation matrix which is based on the $\mathbb{E}_{q(z_i | x^{(n)})}[z_i]$ (called the *conditional means*) instead of samples from $q(z_i | x^{(n)})$. Both are shown in Fig. 20. We can see that the correlation matrices computed based on the conditional means *incorrectly* describe the independence between representations of FactorVAE and $\beta$-VAE. AAE is not affected because it learns deterministic $z_i$ given $x$. Using the correlation matrix is not a principled way to evaluate independence in disentanglement learning.

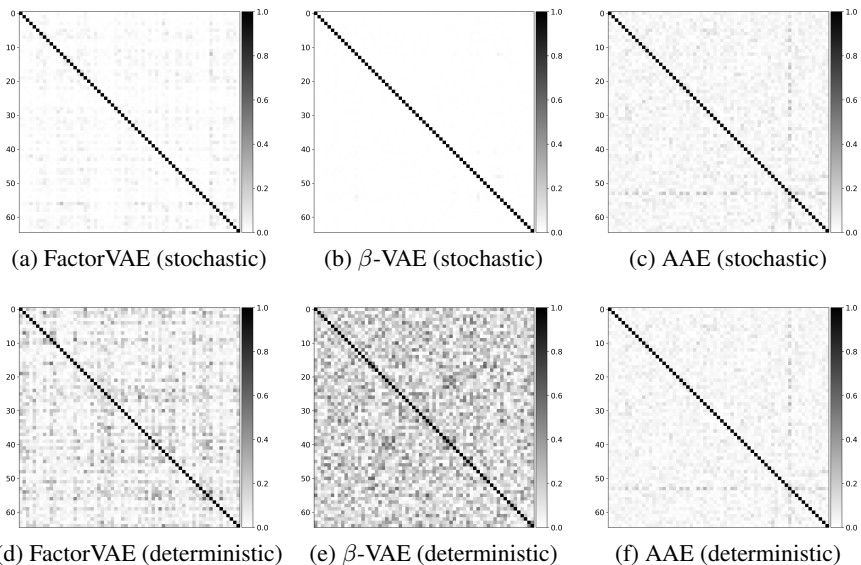

(a) FactorVAE (stochastic)     (b) $\beta$-VAE (stochastic)     (c) AAE (stochastic)

(d) FactorVAE (deterministic)    (e) $\beta$-VAE (deterministic)    (f) AAE (deterministic)

Figure 20: Correlation matrix of representations learned by FactorVAE, $\beta$-VAE and AAE.

### A.7 TRADE-OFF BETWEEN INFORMATIVENESS, INDEPENDENCE AND THE NUMBER OF LATENT VARIABLES

Before starting our discussion, we provide the following fact:

**Fact 2.** *Assume we try to fill a fixed-size pool with fixed-size balls given that all the balls must be inside the pool. The only way to increase the number of the balls without making them overlapped is reducing their size.*

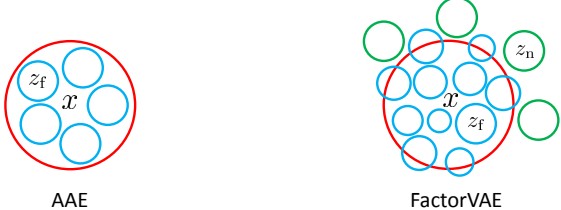

AAE                   FactorVAE

Figure 21: Illustration of representations learned by AAE and FactorVAE. A big red circle represents the total amount of information that $x$ contains or $H(x)$ which is limited by the amount of training data. Blue circles are informative representations $z_\text{f}$ and the size of these circle indicates the informativeness of $z_\text{f}$. Green circles are noisy representations $z_\text{n}$. AAE does not contain $z_\text{n}$, only FactorVAE does.

In the context of representation learning, a pool is $x$ with size $H(x)$ which depends on the training data. Balls are $z_i$ with size $H(z_i)$. Fact. 2 reflects the situation of AAE (see Fig. 21 left). In AAE, all $z_i$ are deterministic given $x$ so the condition "all balls are inside the pool" is met. $H(z_i) \approx$ the entropy of $\mathcal{N}(0, \text{I})$ which is fixed so the condition "fixed-size balls" is also met. Therefore, when the number of latent variables in AAE increases, *all $z_i$ must be less informative* (i.e., $H(z_i)$ must decrease) given that the independent constraint on the latent variables is still satisfied. This is empirically verified in Fig. 22 as we see the distribution of $\mathbb{E}_{q(z_i|x^{(n)})}[z_i]$ over all $x^{(n)} \sim p_{\mathcal{D}}(x)$ becomes narrower when we increase the number of representations from 65 to 200. Also note that increasing the number of latent variable from 65 to 100 does not change the distribution. This suggests that 65 or 100 latent variables are still not enough to capture all information in the data.

FactorVAE, however, handles the increasing number of latent variables in a different way. Thanks to the KL term in the loss function that forces $q(z_i|x)$ to be stochastic, FactorVAE can break the

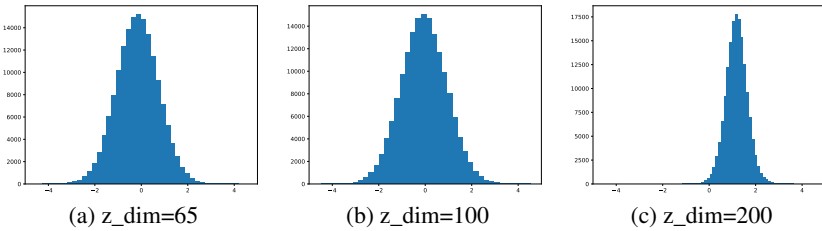

|   (a) z_dim=65   |   (b) z_dim=100   |   (c) z_dim=200   |

Figure 22: Distribution of $\mathbb{E}_{q(z_i|x^{(n)})}[z_i]$ over all $x^{(n)} \sim p_{\mathcal{D}}(x)$ of a particular representation $z_i$ for different AAE models.

constraint in Fact 2 and allows the balls to stay outside the pool (see Fig. 21 right). If we increase the number of latent variables but still enforce the independence constraint on them, FactorVAE will keep a fixed number of informative representations and make all other representations "noisy" with zero informativeness scores. We refer to that capability of FactorVAE as *code compression*.

## A.8 ANALYSIS OF EXISTING METRICS FOR DISENTANGLEMENT

In this section, we analyze recent metrics, including Z-diff score (Higgins et al., 2017a; Kim & Mnih, 2018), Separated Attribute Predictability (SAP) (Kumar et al., 2017), Mutual Information Gap (MIG) (Chen et al., 2018), Disentanglement/Compactness/Informativeness (Eastwood & Williams, 2018), Modularity/Explicitness (Ridgeway & Mozer, 2018).

The main idea behind the Z-diff score (Higgins et al., 2017a; Kim & Mnih, 2018) is that if a ground truth generative factor $y_k$ ($k \in \{0, 1, ..., K\}$) is well aligned with a particular disentangled representation $z_i$ (although we do not know which $i$), we can use a simple classifier to predict $k$ using information from $z$. Higgins et al. (Higgins et al., 2017a) use a linear classifier while Kim et. al. (Kim & Mnih, 2018) use a majority-vote classifier. The main drawback of this metric is that it assumes knowledge about *all* ground truth factors that generate the data. Hence, it is only applicable for a toy dataset like dSprites. Another drawback lies in the complex procedure to compute the metric, which requires training a classifier. Since the classifier is sensitive to the chosen optimizer, hyper-parameters and weight initialization, it is hard to ensure a fair comparison.

The SAP score (Kumar et al., 2017) is computed based on the correlation matrix $C$ between the latent variables $z$ and the ground truth factors $y$. If a latent $z_i$ and a factor $y_k$ are both continuous, the (square) correlation $C_{i,k}$ between them is equal to $\frac{\text{Cov}^2(z_i, y_k)}{\text{Var}(z_k)\text{Var}(y_k)}$ and is in [0, 1]. However, if the factor $y_k$ is discrete, computing the correlation between continuous and discrete variables is not straightforward. The authors handled this problem by learning a classifier that predicts $y_k$ given $z_i$ and used the balanced[8] prediction accuracy as a replacement. Then, for each factor $y_k$, they sorted $C_{:,k}$ in the descending order and computed the difference between the top two scores. The mean of the *difference scores* for all factors was used as the final SAP score. The intuition for this metric is that if a latent $z_i$ is the most representative for a factor $y_k$ (due to the highest correlation score), then other latent variables $z_{\neq i}$ should not be related to $y_k$, and thus, the difference score for $y_k$ should be high. We believe the SAP score is more sensible than Z-diff but it is only suitable when both the ground truth factors and the latent variables are continuous as no classifier is required. Moreover, if we have $K$ discrete ground truth factors and $L$ latent variables, the number of classifiers we need to learn is $L \times K$, which is unmanageable when $L$ is large.

The MIG score (Chen et al., 2018) shares the same intuition as the SAP score but is computed based on the mutual information between every pair of $z_i$ and $y_k$ instead of the correlation coefficient. Thus, the MIG score is theoretically more appealing than the SAP score since it can capture nonlinear relationships between latent variables and factors while the SAP score cannot. The MIG score, to some extent, reflects the concept "interpretability" that we discussed in Section 2 in the main text.

---

[8]To achieve balance, the classifier uses the same number of samples for all categories of $y_k$ during training and testing

Eastwood et. al. (Eastwood & Williams, 2018) proposed three different metrics namely *"disentanglement"*, *"completeness"*, and *"informativeness"* to quantify disentangled representations. These metrics are computed based on a so-called *"important matrix"* $R$ whose element $R_{ik}$ is the relative importance of $z_i$ (w.r.t other $z_{\neq i}$) in predicting $y_k$. More concretely, for each factor $y_k$ ($k = 0, ..., K-1$), they train a regressor (LASSO or Random Forest) to predict $y_k$ from $z$ and use the weight vector provided by this regressor to define $R_{\cdot k}$. The "disentanglement" score $D_i$ quantifies the degree to which a latent $z_i$ captures *at most* one generative factor $y_k$. $D_i$ is computed as $D_i = (1 - H_K(P_{i\cdot}))$ where $H_K(P_{i\cdot}) = \sum_{k=0}^{K-1} -P_{ik} \log P_{ik}$ and $P_{ik} = \frac{R_{ik}}{\sum_{k'=0}^{K-1} R_{ik'}}$ which can be seen as the *"probability"* of predicting $y_k$ instead of $y_{\neq k}$ from $z_i$. Similarly, the "completeness" score $C_k$ quantifies the degree to which a ground truth factor $y_k$ is captured by a single latent $z_i$ ($i = 0, ..., L-1$), computed as $C_k = 1 - H_L(\tilde{P}_{\cdot k})$ where $H_L(\tilde{P}_{\cdot k}) = \sum_{i=0}^{L-1} -\tilde{P}_{ik} \log \tilde{P}_{ik}$ and $\tilde{P}_{ik} = \frac{R_{ik}}{\sum_{i'=0}^{L-1} R_{i'k}}$. The "informativeness" score describes the total amount of information of a particular factor $y_k$ captured by all representations $z$. However, the authors use the prediction error $E_k$ of the $k$-th regressor to quantify "informativeness" instead of $I(y_k, z)$. Despite being well-motivated, these metrics still have several drawbacks. First, using three different metrics to quantify disentangled representations is not as convenient as using a single metric like MIG (Chen et al., 2018). For example, how can we compare two models A and B if A has a better "disentanglement" score but a worse "completeness" score than B? Second, these metrics do not apply for categorical factors with $C$ classes since in this case the model weight is not a vector but an $L \times C$ matrix. Third, defining the pseudo-distribution $P_{ik} = \frac{R_{ik}}{\sum_{k'=0}^{K-1} R_{ik'}}$ seems ad hoc because i) the weight magnitudes $R_{ik}$ are unbounded and can vary significantly (see Appdx. A.9), and ii) $P_{ik}$ strongly depends on the available ground truth factors (e.g. the value of $P_{ik}$ will change if we only consider 2 instead of 5 factors).

Ridgeway et. al. (Ridgeway & Mozer, 2018) proposed two metrics called "modularity" and "explicitness" that have similar interpretations as "disentanglement" and "informativeness" discussed above but differ in implementation. Specifically, they compute the "modularity" score $M_i$ for a representation $z_i$ as $M_i = 1 - \frac{\sum_{k=0}^{K-1}(I(z_i, y_k) - T_{ik})^2}{I^2(z_i, y_{k^*}) \times (K-1)}$ where $k^* = \mathrm{argmax}_k I(z_i, y_k)$ and $T_{ik} = \begin{cases} I(z_i, y_{k^*}) & \text{if } k = k^* \\ 0 & \text{otherwise} \end{cases}$. Like the "disentanglement" score $D_i$, $M_i$ is also ad hoc and is undefined when the number of ground truth factors is 1. The "explicitness" score $E_k$ for each ground truth factor $y_k$ is computed as the ROC curve of a logistic classifier that predicts $y_k$ from $z$. It turns out that $E_k$ is just a way to bypass computing $I(y_k, z)$.

## A.9   THE MUTUAL INFORMATION MATRIX $I(z_i, y_k)$ AND THE IMPORTANCE MATRIX $R_{ik}$

In Fig. 23, we compare our mutual information matrix $I(z_i, y_k)$ with the counterpart in (Ridgeway & Mozer, 2018) and the importance matrix $R_{ik}$ in (Eastwood & Williams, 2018). It is clear that all matrices can capture disentangled representations (those highlighted in red) well since their corresponding values are high compared to other values in the *same column*. However, the matrix $I(z_i, y_k)$ in (Ridgeway & Mozer, 2018) usually overestimates noisy representations since it uses $\mathbb{E}_{q(z_i|x)}[z_i]$ instead of $q(z_i|x)$. The matrix $R_{ik}$ in (Eastwood & Williams, 2018) sometimes assign very high absolute values for noisy representations since the regressor's weights are unbounded. These flaws make the metrics in (Ridgeway & Mozer, 2018) and in (Eastwood & Williams, 2018) inaccurate and unstable, especially "modularity" and "disentanglement" since they require normalization over rows.

## A.10   COMPUTING METRICS FOR INFORMATIVENESS, SEPARABILITY AND INTERPRETABILITY

The metrics for informativeness, separability and interpretability in Section. 3 requires computing $H(z_i)$, $H(z_i|x)$, $H(z_{\neq i})$, $H(z)$, and $H(z_i, y_k)$. We can compute these entropies via quantization or sampling. Quantization is only applicable when $z_i$ is a scalar. If $z_i$ is a high-dimensional vector, we need to use sampling. Below, we describe how to compute $H(z)$ via sampling and $H(z_i)$ via quantization. Other cases can be derived similarly.

```
[[0.6368 0.0056 1.8789 0.0055 0.0056]      [[0.639  0.0056 1.9237 0.0057 0.0053]      [[ 0.0942  0.6402  0.0351  0.0109]
 [0.0001 0.0002 0.     0.0003 0.    ]       [0.2008 0.0377 0.1189 0.1497 0.0243]       [ 0.     43.6432 0.      0.     ]
 [0.     0.0001 0.     0.     0.0003]       [0.1254 0.0946 0.038  0.0262 0.2099]       [ 0.      3.9004 0.      0.     ]
 [0.     0.0002 0.0001 0.0002 0.0001]       [0.146  0.0734 0.0753 0.0591 0.154 ]       [ 1.353  60.784  0.      0.     ]
 [0.     0.0001 0.0003 0.     0.0001]       [0.2576 0.0712 0.1421 0.026  0.0419]       [ 0.     53.6852 1.3094  0.     ]
 [0.0918 0.0412 0.0377 2.7327 0.0253]       [0.072  0.0357 0.0315 2.7142 0.0211]       [ 0.0058  0.5509  8.8005  0.054 ]
 [0.     0.0001 0.     0.0004 0.0001]       [0.1011 0.0537 0.1408 0.193  0.0164]       [ 0.     52.28   0.      0.     ]
 [0.2654 1.0612 0.0404 0.0148 0.0124]       [0.2833 1.0856 0.0649 0.0197 0.019 ]       [ 1.5829  0.4147  0.1026  0.0272]
 [0.0776 0.0162 0.0375 0.0578 2.657 ]       [0.0597 0.013  0.0328 0.044  2.6578]       [ 0.0039  1.1383  0.0004  9.5193]
 [0.2371 0.05   0.1256 0.0041 0.0023]]      [0.2991 0.0614 0.1647 0.0075 0.0044]]      [ 0.5873  1.1907  0.1657  0.0747]]
```

| (a) $I(z_i, y_k)$ w. $q(z_i|x)$ | (b) $I(z_i, y_k)$ w.o. $q(z_i|x)$ | (c) $R_{ik}$ |

Figure 23: **(a):** Our mutual information matrix $I(z_i, y_k)$, **(b):** The mutual information matrix $I(z_i, y_k)$ in (Ridgeway & Mozer, 2018), **(c):** The importance matrix $R_{ik}$ in (Eastwood & Williams, 2018). In (a) and (b), the columns corresponding to the following ground truth factors: "shape", "scale", "rotation", "x-position", "y-position". In (c), the column for "shape" is excluded because the metrics in (Eastwood & Williams, 2018) do not support categorical factors. Values corresponding to disentangled representations are highlighted in red. Defective values are highlighted in green. The model is FactorVAE with TC=20.

**Computing $H(z)$ via sampling**

$$H(z) = -\mathbb{E}_{q(z)}\left[\log q(z)\right]$$

$$= -\mathbb{E}_{q(z,x)}\left[\log \mathbb{E}_{p_{\mathcal{D}}(x)}\left[q(z|x)\right]\right]$$

$$= -\frac{1}{M}\sum_{m=1}^{M}\left[\log \frac{1}{N}\sum_{n=1}^{N} q\left(z^{(m)}|x^{(n)}\right)\right] \tag{13}$$

$$= -\frac{1}{M}\sum_{m=1}^{M}\left[\log \frac{1}{N}\sum_{n=1}^{N}\left(\prod_{i=1}^{L} q\left(z_i^{(m)}|x^{(n)}\right)\right)\right] \tag{14}$$

In Eq. 13, we use Monte Carlo sampling to estimate the expectations outside and inside the log function. The corresponding sample sizes are $M$ and $N$. In Eq. 14, we use the assumption $q\left(z^{(m)}|x^{(n)}\right) = \prod_{i=1}^{L} q\left(z_i^{(m)}|x^{(n)}\right)$. Please note that the entropy $H(z)$ computed via sampling can be negative if $z$ is continuous since we use the density function $q(z|x)$.

**Computing $H(z_i)$ via quantization**    We can compute $H(z_i)$ via quantization as follows:

$$H(z_i) = -\sum_{s_i \in \mathcal{S}} Q(s_i)\log Q(s)$$

where $\mathcal{S}$ is a set of all quantized bins $s_i$ corresponding to $z_i$; $Q(s_i)$ is the probability mass function of $s_i$. To ensure consistency among different $z_i$ as well as different models, we apply the same value range for all latent variables. In practice, we choose the range $[-4, 4]$ since most of the latent values fall within this range. We divide this range into equal-size bins to form $\mathcal{S}$.

We can compute $Q(s_i)$ as follows:

$$Q(s_i) = \frac{1}{N}\sum_{n=1}^{N} Q\left(s_i|x^{(n)}\right)$$

We compute $Q\left(s_i|x^{(n)}\right)$ based on its definition, which is:

$$Q(s_i|x^{(n)}) = \int_a^b q(z_i|x^{(n)})\, dz_i \tag{15}$$

where $a$, $b$ are two ends of the bin $s_i$.

There are two ways to compute $Q(s_i|x^{(n)})$. In the first way, we simply consider the unnormalized $Q'(s_i|x^{(n)})$ as the area of a rectangle whose width is $b - a$ and height is $q(\bar{z}_i|x^{(n)})$ with $\bar{z}_i$ at the center value of the bin $s_i$. Then, we normalize $Q'(s_i|x^{(n)})$ over all bins to get $Q(s_i|x^{(n)})$. In the second way, if $q(z_i|x^{(n)})$ is a Gaussian distribution, we can estimate the above integral with a closed-form function (see Appdx. A.14 for detail).

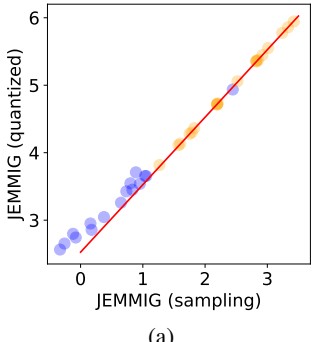 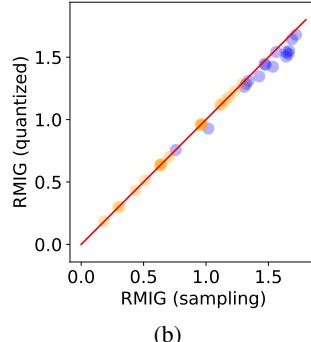

(a)             (b)

Figure 24: Correlation between the sampling (#samples=10000) and quantized (value range=[-4, 4], #bins=100) estimations of JEMMIG/RMIG. In the subplot **(a)**, the red line is $y = x - \log(\text{bin width})$ while in the subplot **(b)**, the red line is $y = x$. Blues denotes FactorVAE models and oranges denotes $\beta$-VAE models. The dataset is dSprites.

### A.11 RELATIONSHIP BETWEEN SAMPLING AND QUANTIZATION

Denote $H_s(z_i|x)$ and $H_q(z_i|x)$ to be the sampling and quantization estimations of an entropy $H(z_i|x)$, respectively. Because $H_s(z_i|x)$ is the expectation of $\log q(z_i|x)$, $H_q(z_i|x)$ is the expectation of $\log Q(z_i|x)$, and $Q(z_i|x) \approx q(z_i|x) \times \text{bin width}$ if the bin width is *small enough*, there exists a *gap* between $H_s(z_i|x)$ and $H_q(z_i|x)$, specified as follows:

$$
\begin{aligned}
H_q(z_i|x) &= H_s(z_i|x) - \log(\text{bin width}) \\
&= H_s(z_i|x) - \log\left(\frac{\text{value range}}{\text{\#bins}}\right) \\
&= H_s(z_i|x) - \log(\text{value range}) + \log(\text{\#bins})
\end{aligned}
$$

Since $Q(z_i) = \mathbb{E}_{p_{\mathcal{D}}(x)}[Q(z_i|x)]$ and $q(z_i) = \mathbb{E}_{p_{\mathcal{D}}(x)}[q(z_i|x)]$, we have $Q(z_i) \approx q(z_i) \times \text{bin width}$. Thus, $H_q(z_i)$ and $H_s(z_i)$ also exhibit a similar gap as $H_s(z_i|x)$ and $H_q(z_i|x)$:

$$
H_q(z_i) = H_s(z_i) - \log(\text{bin width})
$$

However, this gap disappears when computing the mutual information $I(z_i, x)$ since:

$$
\begin{aligned}
I_q(z_i, x) &= H_q(z_i) - H_q(z_i|x) \\
&= (H_s(z_i) - \log(\text{bin width})) - (H_s(z_i|x) - \log(\text{bin width})) \\
&= H_s(z_i) - H_s(z_i|x) \\
&= I_s(z_i, x)
\end{aligned}
$$

In fact, one can easily prove that:

$$
\lim_{\text{\#bins}\to+\infty} I_q(z_i, x) = I_s(z_i, x)
$$

.

Similar relationships between sampling and quantization also apply for $H(z_i, y_k)$ and $I(z_i, y_k)$. They are clearly shown in Fig. 24.

In summary,

- Sampling entropies such as $H_s(z_i|x)$ or $H_s(z_i)$ are usually fixed but *can be negative* since $q(z_i|x)$ or $q(z_i)$ can be $> 1$. However, these entropies *can still be used for ranking* though it is not easy to interpret them.
- Quantized entropies such as $H_q(z_i|x)$ or $H_q(z_i)$ can be positive if the bin width is small enough (or #bins is large enough). The growth rate is $-\log(\text{bin width})$ (or $\log(\text{\#bin})$). Because $\lim_{x\to+\infty} \log x = +\infty$, $H_q(z_i|x)$ and $H_q(z_i)$ *cannot be upper-bounded*.

- The mutual information $I(z_i, x)$ is consistent via either quantization or sampling. Unlike the entropies, $I(z_i, x)$ is well-bounded even when $z_i$ is continuous, thus, is suitable to be used in a metric. However, when #bins is small, the approximation $Q(z_i) \approx q(z_i) \times$ bin width does not hold and quantization estimation can be inaccurate.

## A.12    NORMALIZING JEMMIG

Recall that the formula of the *unnormalized* JEMMIG$(y_k)$ is $H(z_{i^*}, y_k) - I(z_{i^*}, y_k) + I(z_{j^\circ}, y_k)$. If we estimate $H(z_{i^*}, y_k)$ via quantization, the value of the unnormalized JEMMIG$(y_k)$ will vary according to the bin width (or value range and #bins) (as shown in Fig. 17 (left)). However, we can still rank models by forcing them using the same bin width (or the same value range and #bins). To avoid setting these hyper-parameters, we can estimate $H(z_{i^*}, y)$ via sampling. In this case, the value of the unnormalized JEMMIG$(y_k)$ only depends on $q(z_i|y)$ which is fixed after learning. Ranking disentanglement models using the unnormalized JEMMIG$(y_k)$ is somewhat similar to *ranking generative models using the log-likelihood*.

Using the unnormalized JEMMIG$(y_k)$ causes interpretation difficulty. We could normalize JEMMIG$(y_k)$ as follows:

$$\frac{H_{\mathrm{q}}(z_i) + H(y_k) - 2I(z_{i^*}, y_k) + I(z_{j^\circ}, y_k)}{H_{\mathrm{q}}(u) + H(y_k)} \tag{16}$$

where $H_{\mathrm{q}}(z_i)$ is a quantization estimation of $H(z_i)$, hence, greater than 0; $H_{\mathrm{q}}(u)$ is an entropy that bounds $H_{\mathrm{q}}(z_i)$ but does not depend on $q(z_i|x)$. Intuitively, $u$ should be uniform. The main problem is how to find an effective value range $[a, b]$ of $z_i$ that satisfies 2 conditions: i) most of the mass of $z_i$ falls within that range, and ii) $H(u)$ is the bound of $H(z_i)$ if $u \in [a, b]$. However, before solving this question, we try to answer a similar yet easier question: "Given a Gaussian random variable $z \sim \mathcal{N}(\mu, \sigma)$, what is the value range of a uniform random variable $u$ such that $H(u) \geq H(z)$?". Assume $u \in [a, b]$, the entropy of $u$ is $H(u) = \log(b - a)$ while the entropy of $z$ is $H(z) = 0.5 \log(2\pi e\sigma^2)$. We have:

$$H(z) \leq H(u)$$
$$\Leftrightarrow 0.5 \log(2\pi e\sigma^2) \leq \log(b - a)$$
$$\Leftrightarrow \sigma\sqrt{2\pi e} \leq b - a$$

Thus, to ensure $H(u)$ to be an upper bound of $H(z)$, we should choose the value range of $u$ to be at least $\sigma\sqrt{2\pi e}$. If $\sigma = 1$, this range is about $4.1327$. If we also want $[a, b]$ to capture most of the mass of $z$, $a$ should be $\mu - \frac{\sigma}{2}\sqrt{2\pi e}$ and $b$ should be $\mu + \frac{\sigma}{2}\sqrt{2\pi e}$.

Come back to the main problem, since $q(z_i) = \mathbb{E}_{p_\mathcal{D}(x)}[q(z_i|x)]$ and $q(z_i|x)$ is usually a Gaussian distribution $\mathcal{N}(\mu_i, \sigma_i)$, we can choose $a, b$ as follows:

$$a = \min\left(\mu_i^{(1)} - \frac{\sigma_i^{(1)}}{2}\sqrt{2\pi e}, ..., \mu_i^{(N)} - \frac{\sigma_i^{(N)}}{2}\sqrt{2\pi e}\right), \text{ and}$$

$$b = \max\left(\mu_i^{(1)} + \frac{\sigma_i^{(1)}}{2}\sqrt{2\pi e}, ..., \mu_i^{(N)} + \frac{\sigma_i^{(N)}}{2}\sqrt{2\pi e}\right)$$

One may wonder that different methods can choose different value ranges $[a, b]$ to normalize JEMMIG so how to ensure a fair comparison among them using the normalized JEMMIG. A simple solution is using the same value range $[a, b]$ for different models. In this case, $b - a$ should be large enough to cover various distributions. We can write Eq. 16 as follows:

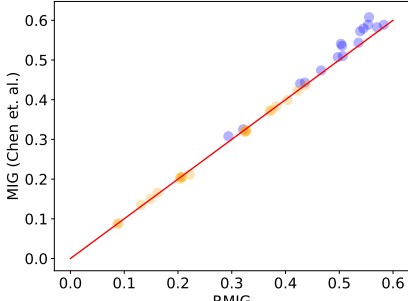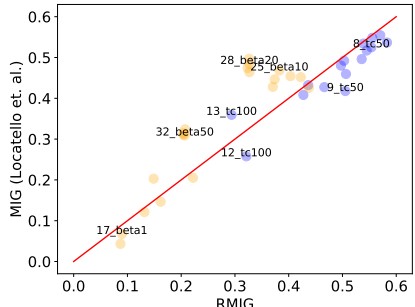

Figure 25: **Left:** Correlation between our RMIG (#bins=100) and the original MIG (Chen et al., 2018) (#samples=10000). **Right:** Correlation between our RMIG (#bins=100) and the implementation of MIG in (Locatello et al., 2019) (#bins=100). Experiments are conducted on the dSprites dataset.

$$\frac{H_q(z_i) + H(y_k) - 2I(z_{i^*}, y_k) + I(z_{j^\circ}, y_k)}{H_q(u) + H(y_k)}$$
$$= \frac{H_s(z_i) - \log(\text{value range}) + \log(\#\text{bins}) + H(y_k) - 2I(z_{i^*}, y_k) + I(z_{j^\circ}, y_k)}{H_s(u) - \log(\text{value range}) + \log(\#\text{bins}) + H(y_k)}$$
$$= \frac{H_s(z_i) - \log(\text{value range}) + \log(\#\text{bins}) + H(y_k) - 2I(z_{i^*}, y_k) + I(z_{j^\circ}, y_k)}{\log(\#\text{bins}) + H(y_k)} \quad (17)$$

Since the fraction in Eq. 17 is smaller than 1, increasing #bins will increase this fraction but still ensure that it is smaller than 1. This means the normalized JEMMIG is always in $[0, 1]$ despite #bins.

### A.13 COMPARING RMIG WITH OTHER MIG IMPLEMENTATIONS

RMIG has several advantages compared to the original MIG (Chen et al., 2018) which we refer as MIG1: i) RMIG works on real datasets, MIG1 does not; ii) RMIG supports continuous factors, MIG1 does not. On toy datasets such as dSprites, RMIG produces almost the same results as MIG1 (Fig. 25 (left)). We argue that the small differences between RMIG and MIG1 scores in some models are caused by either the quantization error of RMIG (when #bins=100) or the sampling error of MIG1 (when #samples=10000).

Locatello et. al. (Locatello et al., 2019) provided an implementation[9] of MIG which we refer as MIG2. MIG2 is *theoretically incorrect* in two points: i) it only uses the mean of the distribution $q(z_i|x^{(n)})$ instead of the whole distribution $q(z_i|x^{(n)})$, and ii) the bin range and width varies for different $z_i$. The performance of MIG2 is, thus, unstable. We can easily see this problem by comparing the right plot with the left plot in Fig. 25. MIG2 usually overestimates the true MIG1 when evaluating $\beta$-VAE models with a large $\beta$ (e.g. $\beta \in \{20, 30, 50\}$). We guess the reason is that in these models, $q(z_i|x^{(n)})$ usually has high variance, hence, using the mean of $q(z_i|x^{(n)})$ like MIG2 leads to the wrong estimation of $I(z_i, y_k)$.

### A.14 DEFINITE INTEGRAL OF A GAUSSIAN DENSITY FUNCTION

Assume that we have a Gaussian distribution $\mathcal{N}(\mu, \sigma)$. The definite integral of its density function within the range $[a, b]$ denoted as $G(a, b)$ can be computed as follows:

$$
\begin{aligned}
G(a, b) &= \int_a^b \frac{1}{\sigma\sqrt{2\pi}} \exp\left(\frac{-(x-\mu)^2}{2\sigma^2}\right) dx \\
&= \frac{1}{2}\left(\text{erf}\left(\frac{b-\mu}{\sigma\sqrt{2}}\right) - \text{erf}\left(\frac{a-\mu}{\sigma\sqrt{2}}\right)\right)
\end{aligned}
$$

---

[9]https://github.com/google-research/disentanglement_lib

Although $\text{erf}(\cdot)$ does not have analytical form, we can compute its values with high precision using polynomial approximation. For example, the following approximation provides a maximum error of $5 \times 10^{-4}$ (Def, 2019):

$$\text{erf}(x) \approx 1 - \frac{1}{(1 + a_1 x + a_2 x^2 + a_3 x^3 + a_4 x^4)^4}, \ x > 0$$

where $a_1 = 0.278393$, $a_2 = 0.230389$, $a_3 = 0.000972$, $a4 = 0.078108$.

### A.15 REPRESENTATIONS LEARNED BY FACTORVAE

We empirically observed that FactorVAE learns the same set of disentangled representations across different runs with varying numbers of latent variables (see Appdx. A.18). This behavior is akin to that of deterministic PCA which uncovers a fixed set of linearly independent factors[10] (or principal components). Standard VAE is theoretically similar to probabilistic PCA (pPCA) (Tipping & Bishop, 1999) as both assume the same generative process $p(x, z) = p_\theta(x|z)p(z)$. Unlike deterministic PCA, pPCA learns a rotation-invariant family of factors instead of an identifiable set of factors. However, in a particular pPCA model, the relative orthogonality among factors is still preserved. This means that the factors learned by different pPCA models are statistically equivalent. We hypothesize that by enforcing independence among latent variables, FactorVAE can also learn statistically equivalent factors (or $q(z_i|x)$) which correspond to visually similar results. We provide a proof sketch for the hypothesis in Appdx. A.16. We note that Rolinek et. al. (Rolinek et al., 2018) also discovered the same phenomenon in $\beta$-VAE.

### A.16 WHY FACTORVAE CAN LEARN CONSISTENT REPRESENTATIONS?

Inspired by the variational information bottleneck theory (Alemi et al., 2016), we rewrite the standard VAE objective in an equivalent form as follows:

$$\min_{q(z|x)} I(x, z) \quad \text{s.t.} \quad \text{Rec}(x) \leq \beta \tag{18}$$

where $\text{Rec}(x)$ denotes the reconstruction loss over $x$ and $\beta$ is a scalar.

In the case of FactorVAE, since all latent representations are independent, we can decompose $I(x, z)$ into $\sum_i I(x, z_i)$. Thus, we argue that FactorVAE optimizes the following information bottleneck objective:

$$\min_{q(z|x)} \sum_i I(x, z_i) \quad \text{s.t.} \quad \text{Rec}(x) \leq \beta \tag{19}$$

We assume that $\text{Rec}(x)$ represents a fixed condition on all $q_i(z|x)$. Because $I(x, z_i)$ is a convex function of $q(z_i|x)$ (see Appdx. A.17), minimizing Eq. 19 leads to unique solutions for all $q(z_i|x)$ (Note that we do not count permutation invariance among $z_i$ here).

To make $\text{Rec}(x)$ a fixed condition on all $q_i(z|x)$, we can further optimize $p(x|z)$ with $z$ sampled from a fixed distribution like $\mathcal{N}(0, \text{I})$. This suggests that we can add a GAN objective to the original FactorVAE objective to achieve more consistent representations.

### A.17 $I(x, z)$ IS A CONVEX FUNCTION OF $p(z|x)$

Let us first start with the definition of a convex function and some of its known properties.

**Definition 3.** Let $X$ be a set in the real vector space $\mathbb{R}^D$ and $f : X \to \mathbb{R}$ be a function that output a scalar. $f$ is **convex** if $\forall x_1, x_2 \in X$ and $\forall \lambda \in [0, 1]$, we have:

$$f(\lambda x_1 + (1 - \lambda)x_2) \leq \lambda f(x_1) + (1 - \lambda)f(x_2)$$

**Proposition 4.** *A twice differentiable function $f$ is **convex** on an interval if and only its second derivative is **non-negative** there.*

---

[10]When we mention factors in this context, they are not really factors of variation. They refer to the columns of the projection matrix W in case of PCA and the component encoding functions $q(z_i|x)$ in case of deep generative models.

**Proposition 5** (Jensen's inequality). *Let $x_1, ..., x_n$ be real numbers and let $a_1, ..., a_n$ be positive weights on $x_1, ..., x_n$ such that $\sum_i^n a_i = 1$. If $f$ is a convex function on the domain of $x_1, ..., x_n$, then*

$$f\left(\sum_{i=1}^n a_i x_i\right) \leq \sum_{i=1}^n a_i f(x_i)$$

*Equality holds if and only if all $x_i$ are equal or $f$ is a linear function.*

**Proposition 6** (Log-sum inequality). *Let $a_1, ..., a_n$ and $b_1, ..., b_n$ be non-negative numbers. Denote $a = \sum_{i=1}^n a_i$ and $b = \sum_{i=1}^n b_i$. We have:*

$$\sum_{i=1}^n a_i \log \frac{a_i}{b_i} \geq a \log \frac{a}{b}$$

*with equality if and only if $\frac{a_i}{b_i}$ are equal for all $i$.*

Armed with the definition and propositions, we can now prove that $I(x, z)$ is a convex function of $p(z|x)$. Let $p_1(z|x)$ and $p_2(z|x)$ be two distributions and let $p_\star(z|x) = \lambda p_1(z|x) + (1 - \lambda)p_2(z|x)$ with $\lambda \in [0, 1]$. $p_\star(z|x)$ is a valid distribution since $p_\star(z|x) > 0 \ \forall z$ and $\int_x \int_z p_\star(z|x)p(x) \, dz \, dx = 1$. In addition, we have:

$$
\begin{aligned}
p_\star(z) &= \int_x p_\star(z|x)p(x) \, dx \\
&= \int_x \left(\lambda p_1(z|x) + (1 - \lambda)p_2(z|x)\right) p(x) \, dx \\
&= \lambda \int_x p_1(z|x)p(x) \, dx + (1 - \lambda) \int_x p_2(z|x)p(x) \, dx \\
&= \lambda p_1(z) + (1 - \lambda)p_2(z)
\end{aligned}
$$

We write $I(x, z) = \lambda I_1(x, z) + (1 - \lambda)I_2(x, z)$ as follows:

$$
\begin{aligned}
I(x, z) =& \lambda \int_x p(x) \int_z p_1(z|x) \log \frac{p_1(z|x)}{p_1(z)} \, dz \, dx + \\
&+ (1 - \lambda) \int_x p(x) \int_z p_2(z|x) \log \frac{p_2(z|x)}{p_2(z)} \, dz \, dx \\
=& \int_x p(x) \int_z \left( \lambda p_1(z|x) \log \frac{\lambda p_1(z|x)}{\lambda p_1(z)} + (1 - \lambda)p_2(z|x) \log \frac{(1 - \lambda)p_2(z|x)}{(1 - \lambda)p_2(z|x)} \right) \, dz \, dx \\
\geq& \int_x p(x) \int_z p_\star(z|x) \log \frac{p_\star(z|x)}{p_\star(z)} \, dz \, dx \qquad (20) \\
=& I_\star(x, z)
\end{aligned}
$$

where the inequality in Eq. 20 is the log-sum inequality. This completes the proof.

## A.18 EXPERIMENTS TO SHOW THAT FACTORVAE LEARNS CONSISTENT REPRESENTATIONS

We first trained several FactorVAE models with 3 latent variables on the CelebA dataset. After training, for each model, we performed 2D interpolation on every pair of latent variables $z_i$, $z_j$ ($i \leq j$) and decoded the interpolated latent representations back to images for visualization. We found that the learned representations from these models share *visually similar patterns*, which is illustrated in Fig. 26. It is apparent that all images in Fig. 26 are derived from a single one (e.g. we can choose the first image as a reference) by switching the rows and columns and/or flipping the whole image vertically/horizontally. The reason why switching happens is that all latent variables of FactorVAE are permutation invariant. Flipping happens due to the symmetry of $q(z_i)$ which is forced to be similar to $p(z_i) = \mathcal{N}(0, \mathrm{I})$.

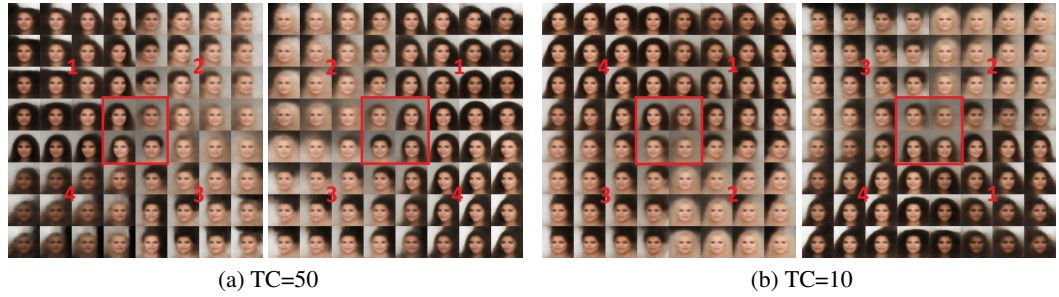

(a) TC=50                                    (b) TC=10

Figure 26: Random traversal on the latent space of FactorVAE. We can easily see the visual resemblance among image regions corresponding the same number.



(a) TC=10, z_dim=65      (b) TC=50, z_dim=65      (c) TC=50, z_dim=100      (d) TC=50, z_dim=200

Figure 27: Top 10 representations sorted by the variance of the distribution of $\mathbb{E}_{q(z_i|x^{(n)})}[z_i]$ over all $x^{(n)}$.

We then repeated the above experiment on FactorVAE models containing 65, 100, 200 latent variables, but replacing 2D interpolation on pairs of latent variables with conditional 1D interpolation on individual latent variables to account for large numbers of combinations. We sorted the latent variables $z_i$ of each model according to the variance of the distribution of $\mathbb{E}_{q(z_i|x^{(n)})}[z_i]$ over all data samples $x^{(n)} \sim p_{\mathcal{D}}(x)$ in descending order. Fig. 27 shows results for the top 10 latent variables (of each model). We can see that some factors of variation are *consistently learned* by these models, for example, those that represent changes in color of the image background. Because these factors usually appear on top, we hypothesize that the learned factors should follow *some fixed order*. However, many pronounced factors do not appear at the top, suggesting that the sorting criterion is inadequate. We then used the informativeness metric defined in Sec. 4.1 to sort the latent variables. Now the "visual consistency" and "ordering consistency" patterns emerge, (see Fig. 28). We also observed that the number of learned factors is relatively fixed (around 38-43) for all models despite that the number of latent variables varies significantly from 65 to 200.

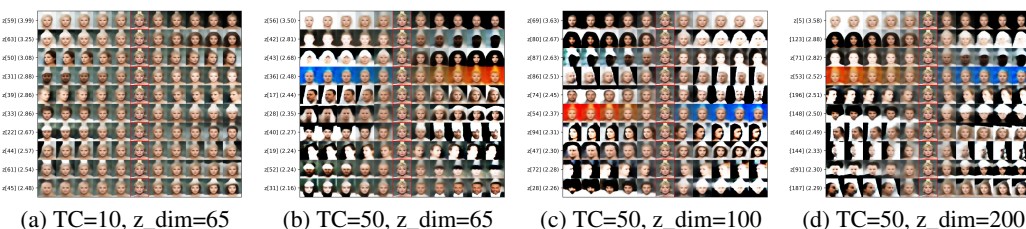

(a) TC=10, z_dim=65      (b) TC=50, z_dim=65      (c) TC=50, z_dim=100      (d) TC=50, z_dim=200

Figure 28: Top 10 representations sorted by informativeness scores. We can clearly see the consistency of representations across different runs.

