# OpenReview forum: "Theory and Evaluation Metrics for Learning Disentangled Representations"
_ICLR.cc/2020/Conference — Accept (Poster)_

### Official Review · AnonReviewer3 · 2019-10-19
**Official Blind Review #3**

**Rating:** 6

**Review:**

This paper defines precise semantics of disentanglement representations and presents evaluation metrics to evaluate such representations. The authors provides information-theoretic characterization disentangled representations along three dimensions: informativeness, separability, and interpretability; and propose metrics to measure them. The authors argue that:- Informativeness can be defined as the mutual information between a particular representation (or a group of representations) w.r.t the data---I(x,z_i) - Separability between two representations can be achieved if they do not share any common information about the data---I(x,z_i,z_j) = 0 - Interpretability with respect to a concept is achieved if a representation contains information about the concept---I(z_i, y_k) = H(z_i) = H(y_k).The authors define a disentangled representation as a representation that is fully separable and fully interpretable and propose a suite of metrics to evaluate disentangled representations.Finally, the authors evaluate several representation learning methods (FactorVAE, betaVAE, AAE) using these metrics on toy and real datasets.

I think this paper addresses an important problem of quantifying and measuring disentangled representations. The proposed metrics are reasonably sound and the authors provide an extensive set of experiments to show how to use them in practice.

I have one comment regarding the sheer number of metrics that are presented in the paper and their practical usage. How do the authors see them being used in the future to compare models? Is one of the main arguments of the paper to encourage other people to use all of these metrics? Are all of them needed, given some of them seem to correlate with others? I think it would be helpful to highlight a few metrics or aggregates of them to inform future research in this area.

Also, for the paper to be more self-contained, I think the authors should include a short discussion about models that they compare in the experiments.


**Experience Assessment:**

I have published one or two papers in this area.

**Review Assessment: Checking Correctness Of Derivations And Theory:**

I assessed the sensibility of the derivations and theory.

**Review Assessment: Checking Correctness Of Experiments:**

I assessed the sensibility of the experiments.

**Review Assessment: Thoroughness In Paper Reading:**

I read the paper at least twice and used my best judgement in assessing the paper.

---

> ### Author Response · Authors · 2019-11-13
> **Response to Reviewer 3**
>
> Thank you for your insightful comments.
>
> 1) >> I have one comment regarding the sheer number of metrics …
> In Table 1 and in the experiment part, we theoretically and empirically show that our proposed metrics are more advanced than the existing metrics for comparing disentanglement learning models. We hope that with these advantages, our metrics will be widely used in the future.
>
> The reason we proposed several correlated metrics (e.g. WSPEIN, SEPIN@k, WINDIN, INDIN@k or RMIG, JEMMIG) in our paper is that we want to make our work as comprehensive as possible by considering different scenarios and approaches when designing our metrics.
>
> We agree with Reviewer 3 that we should highlight some key metrics to inform future research. Based on our definition of disentangled representations, JEMMIG and WSEPIN are the two keys metrics in case ground truth factors are and are not available, respectively. We have added this point to the Discussion part in our revised version.
>
> 2) >> Also, for the paper to be more self-contained, I think the authors should include a short discussion about models that they compare in the experiments.
> We thank Reviewer 3 for this useful advice. We have added a discussion about these models in the Appdx. 1 of the revised version.

---

### Official Review · AnonReviewer2 · 2019-10-22
**Official Blind Review #2**

**Rating:** 6

**Review:**

This paper provides a mathematically-grounded set of axes to describe the effectiveness of latent representations: informativeness (as the mutual information between input and z), separability (I(x, zi, zj) = 0, i!=j), and interpretability. They accompany these with metrics to evaluate representations across those criteria.

They use this to evaluate B-VAE, FactorVAE and AAE. Tl;dr B-VAE are the most separable, FactorVAE are the most interpretable.

I think the paper serves as a great primer for people who are not familiar with disentangled representations, and also proposes a necessary vocabulary for understanding the trade-offs of different representation disentangling methods.

The paper is too long, you could cut the final paragraph of S2.1 without losing anything (that's half a page already). You can (and should) edit this down. I don't think this paper should be longer than 8 pages.

P.s. did you really cite "Error Function" to the wikipedia page for Error function?

**Experience Assessment:**

I do not know much about this area.

**Review Assessment: Checking Correctness Of Derivations And Theory:**

I assessed the sensibility of the derivations and theory.

**Review Assessment: Checking Correctness Of Experiments:**

I assessed the sensibility of the experiments.

**Review Assessment: Thoroughness In Paper Reading:**

I made a quick assessment of this paper.

---

> ### Author Response · Authors · 2019-11-13
> **Response to Reviewer 2**
>
> Thank you for your comments.
>
> We agree that our paper (including the appendix) is long because we tried to make the paper as comprehensive and self-contained as possible. We hope that Reviewer 2 (and other readers) will find our paper enjoyable to read.
>
> For the reason of comprehensiveness, we believe the two paragraphs in Section 2.1 are important to place our concepts in context and highlight the difference between our work and other related works such as [1, 2, 3]. We hope the readers will benefit from being informed, as suggested by Reviewer 1 and Reviewer 3. Therefore, we decide to keep them in our revised version.
>
> We would like to say that we do not cite the “error function” but its approximations that give low errors.
>
> [1] Towards a definition of disentangled representations, Higgins et. al., 2018
> [2] A framework for the quantitative evaluation of disentangled representations, Eastwood&Williams, 2018
> [3] Learning deep disentangled embeddings with the f-statistic loss, Ridgeway et. al., 2018

---

### Official Review · AnonReviewer1 · 2019-10-26
**Official Blind Review #1**

**Rating:** 6

**Review:**

Summary:

The paper presents a new set of metrics for evaluating disentangled representations in both supervised and unsupervised settings. Disentangled representations are evaluated along three dimensions: informativeness, separability, and interpretability. While previous work offers metrics for similar dimensions (e.g., (Eastwood & Williams, 2018)), the paper suggests that the metrics of the submission are superior to (Eastwood & Williams, 2018), based on a comparison between FactorVAE and Beta-VAE.

Strengths:

(1) The metrics of the paper are well motivated from an information-theoretic standpoint. While, in some cases, the metrics themselves are straight applications of information theory (e.g., informativeness metric == mutual information), the authors came up with new metrics when existing information-theoretic definitions could be fooled by, e.g., introducing noisy and independent latent representations z.

(2) Experimental results show that these metrics are better than previous metrics at discriminating between FactorVAE and Beta-VAE (in an experimental setting where the former is clearly superior to the latter).

(3) As opposed to much of prior work, these metrics do not require any training, which can be considered a plus as they do not require adaptation for each (sub)domain.

(4) The paper is overall well written and clear. I very much enjoyed reading it.

Weaknesses:

(1) My main concern with the paper, which makes me vote for “weak accept” instead of “accept”: The metrics of the paper are compared against previous metrics (Eastwood & Williams, 2018) on *only two* types of disentangled representations, namely FactorVAE and Beta-VAE, and in one experimental condition. (There is plenty of other experimental material in the appendix which also introduces AAE, but none of it seems to compare against previous *metrics*). I think comparing metrics on only two systems is somewhat problematic, and we don’t know how general the results are. I would have preferred to see FactorVAE and Beta-VAE evaluated with fewer hyperparameter choices to allow introducing more variational approaches. Could you (the authors) at least evaluate AAE against Eastwood & Williams too? This concern is not just mere quibbling, as the authors have shown (e.g., in Section 3.2) that specific edge cases can fool naïve metrics (e.g., by introducing noisy and independent latent variables), and there may be other edge cases that the authors have not considered. I think evaluating new metrics against previous work (i.e., metrics) on only two underlying systems is not very convincing. I acknowledge that the paper offers *lots* of experiments – the full pdf is 30 pages (!) – but I think some of the existing ones could go to leave space for evaluations using more underlying VAE/baselines/edge-case systems.

(2) The paper presents six metrics (MI, MISJED, WSEPIN, WINDIN, RMIG, JEMMIG) along three dimensions. While each individual metric makes sense, I feel the paper lacks a discussion section that ties these pieces together and suggests a way of using these metrics conjointly across the three dimensions towards building better-disentangled representations (the paper has a “discussion” section, but it is very short and actually more of a conclusion). The more metrics we have, the more chances each underlying VAE model can “win” on one of the metrics, which is not particularly enlightening.

(3) The paper is not self-contained, and some parts are almost impossible to understand without familiarity with (Kim & Mnih, 2018) and (Higgins et al., 2017a). It uses technical terms of these papers without explanations (e.g., TC is not even spelled out it seems). Hyperparameters of these papers (e.g., beta, gamma) are used without explanation.

(4) There is no related work section. Such a section could, e.g., make what is borrowed from (Kim & Mnih, 2018; Higgins et al., 2017a) more understandable.

Overall, I think it is a nice paper that makes significant contributions to the problem of building better disentangles representations thanks to better evaluation metrics. The empirical support for some of the claims (e.g., superiority to (Eastwood & Williams, 2018)) is a bit weak, but other strengths mentioned above largely make up for that.

Minor comments:

The definitions of SEPIN@k and INDIN@k don’t seem quite right. The summation iterates over z_0, …, z_k-1, leaving off z_k, …, z_L-1, but the latter variables might contain some z’s with lowest mutual information with x. The way the ‘sorted’ function is written only has the effect of reordering z_0, …, z_{k-1} among themselves, never considering any of the z_k and above whatever their MI’s are, which is probably not what the authors meant. Perhaps ‘sorted’ is intended to both sort and rename z’s, but if z’s are indeed renamed I think this should be indicated mathematically otherwise the equation is wrong (e.g., (z’_1, …, z’_{L-1}) = sorted_by_MI(z_1, …, z_{L_1})).

Typos in WSEPIN and WINDIN definitions: “0 = 1” -> “i = 1”?

Figure 1: “Image”? The paper is written in general terms and doesn’t seem to assume otherwise x is an image representation (or does it?).

Section 3.4: “Table. 1” -> “Table 1”

Table 1 vs. title of Section 3.1: The author’s informativeness metric doesn’t have a name, but the section title contains “informativeness,” which could be confusing vs. “informativeness” in Table 1 which is a completely different metric.

Page 8, interpretability: “TC=10”. Do you mean “TC loss” here? I presume 10 is the value of a hyperparameter of the FactorVAE paper (gamma?), and not the actual value of the TC term. Same question for Figure 9 and later figures of the appendix.

**Experience Assessment:**

I have published in this field for several years.

**Review Assessment: Checking Correctness Of Derivations And Theory:**

I carefully checked the derivations and theory.

**Review Assessment: Checking Correctness Of Experiments:**

I carefully checked the experiments.

**Review Assessment: Thoroughness In Paper Reading:**

I read the paper thoroughly.

---

> ### Author Response · Authors · 2019-11-13
> **Response to Reviewer 1**
>
> Thank you for your detailed comments and suggestions. We would like to address your concerns as follows:
>
> 1) >> My main concern with the paper, …
> The main reason for focusing on FactorVAEs and BetaVAEs is that the two models clearly highlight the difference between metrics, and these justify the need for better metrics, which motivated our research in the first place. As shown in Figs. 5a,b and Figs. 7a,b, RMIG and JEMMIG favor FactorVAEs to BetaVAEs while “Disentanglement” and “Completeness” favor BetaVAEs over FactorVAEs. Such contradiction between two evaluation systems allows us to confidently infer that our metrics are more accurate than those in [1] if we can show that FactorVAEs actually learn better disentangled representations than BetaVAEs. In our paper, we verify this hypothesis using visualization (Fig. 4 and Fig. 16), and also draw from the support by the work of Kim et. al., 2018 [2].
>
> The AAE models, on the other hand, do not reveal differences between metrics. Both ours and those metrics in [1] show that AAEs are not as good as BetaVAEs but with different gaps (https://ibb.co/7NxtTvQ (JEMMIG), https://ibb.co/g9hKKYJ (RMIG), https://ibb.co/gS4fbtm (Disentanglement), https://ibb.co/PWL83Mm (Completeness)). Visualization also provides little help because neither BetaVAEs nor AAEs show good interpretability.
>
> Please note that the experiments on FactorVAEs and BetaVAEs are mainly to show that our metrics produce more accurate and more reasonable results than the metrics in [1]. There are other advantages of our metrics that these metrics do not have. For example, our metrics are theoretically sound, our metrics do not use classifiers, our metrics produce a single score (compared to 3 different scores of [1]). We refer Reviewer 1 to Table 1, Appdx. 7, Appdx. 8 for detailed analyses.
>
> However, we agree with the Reviewer 1 that our work has not covered all possible edge cases and there may exist situations in which our metrics can be fooled (and it does not mean that existing metrics cannot be fooled in these cases). We leave this room for future exploration and hope that other researchers can improve upon our work to derive better metrics for disentanglement learning.
>
> 2) >> The paper presents six metrics …
> We thank Reviewer 1 for pointing that out. We have revised the Discussion section to highlight WSEPIN and JEMMIG as our key metrics.
>
> 3) >> The paper is not self-contained, and some parts are almost impossible to understand without …
> We acknowledge that we assume readers are familiar with FactorVAEs, BetaVAEs, AAEs before reading our paper. We have added detailed description of these models in the Appdx. 1 in the revised version.
>
> >>Some hyperparameters (TC, Beta) were used without explanation…
> In the revised version, we have added a sentence to the second paragraph in the Experiment section to explicitly describe these hyperparameters.
>
> 4) >> There is no related work section. Such a section could, e.g., make what is borrowed from (Kim & Mnih, 2018; Higgins et al., 2017a) more understandable.
> Thank you for pointing that out. Since the main text is already quite long (Reviewer 2 even advises to cut 2 pages down!), we chose to embed related work in Introduction and focus on the metrics. More detailed comparisons between our metrics and the existing work are included in Appdx. 8: Analysis of existing metrics for disentanglement.
>
> 5) >> The definitions of SEPIN@k and INDIN@k don’t seem quite right…
> In case of SEPIN@k and INDIN@k, we want to do both sorting and re-indexing. We have modified our notations in the revised version to resolve the ambiguity pointed out by Reviewer 1.
>
> 6) >> Page 8, interpretability: “TC=10”. Do you …
> ‘TC’ here refers to the coefficient of the additional TC loss in the FactorVAE model. It is the gamma hyperparameter in the original FactorVAE paper (Kim et. al., 2018)
>
> 7) Finally, we thank Reviewer 1 for pointing out typos in our paper and we have fixed all of them in our revised version.
>
> References
> ==========
> [1] A framework for the quantitative evaluation of disentangled representations, Eastwood and Williams, 2018
> [2] Disentangling by factorizing, Kim et. al., 2018

---

### Author Response · Authors · 2021-02-13
**Code for this work**

The code for our work can be found here: https://github.com/clarken92/DisentanglementMetrics

---

### Decision · Program_Chairs · 2019-12-19

**Decision:**

Accept (Poster)

**Comment:**

This manuscript proposes and evaluates new metrics for measuring the quality of disentangled representations for both supervised and unsupervised settings. The contributions include conceptual definitions and empirical evaluation.

In reviews and discussion, the reviewers and AC note missing or inadequate empirical evaluation with many available methods for learning disentangled representations. On the writing, reviewers mentioned that the conciseness of the manuscript could be improved. The reviewers also mentioned incomplete references and discussion of prior work, which should be improved.